# A new class of cyclin dependent kinase in Chlamydomonas is required for coupling cell size to cell division

**Yubing Li[1,2†], Dianyi Liu[3†], Cristina López-Paz[1,3], Bradley JSC Olson[1‡], James G Umen[1,3*]**

[1]Plant Biology Laboratory, The Salk Institute for Biological Studies, La Jolla, United States; [2]Plant Molecular and Cell Biology Program, the Horticultural and Plant Science Department, University of Florida, Gainesville, United States; [3]Donald Danforth Plant Science Center, St. Louis, United States

**Abstract** Proliferating cells actively control their size by mechanisms that are poorly understood. The unicellular green alga *Chlamydomonas reinhardtii* divides by multiple fission, wherein a 'counting' mechanism couples mother cell-size to cell division number allowing production of uniform-sized daughters. We identified a sizer protein, CDKG1, that acts through the retinoblastoma (RB) tumor suppressor pathway as a D-cyclin-dependent RB kinase to regulate mitotic counting. Loss of CDKG1 leads to fewer mitotic divisions and large daughters, while mis-expression of CDKG1 causes supernumerous mitotic divisions and small daughters. The concentration of nuclear-localized CDKG1 in pre-mitotic cells is set by mother cell size, and its progressive dilution and degradation with each round of cell division may provide a link between mother cell-size and mitotic division number. Cell-size-dependent accumulation of limiting cell cycle regulators such as CDKG1 is a potentially general mechanism for size control.

*For correspondence: jumen@danforthcenter.org

[†]These authors contributed equally to this work

Present address: [‡]Molecular, Cellular and Developmental Biology Program, the Ecological Genomics Institute, The Division of Biology, Kansas State University, Manhattan, United States

Competing interests: The authors declare that no competing interests exist.

## Introduction

Active maintenance of cell size has been observed across eukaryotic taxa (*Jorgensen and Tyers, 2004*; *Tzur et al., 2009*; *Umen, 2005*), yet the molecular bases for eukaryotic cell-size homeostasis have not been extensively investigated outside of fungi. Cell-size checkpoints were first described in budding and fission yeasts (*Fantes, 1977*; *Johnston et al., 1977*) and subsequently investigated using a variety of approaches (*Jorgensen and Tyers, 2004*; *Turner et al., 2012*). Bistable switches that control cell cycle transitions are increasingly well understood (*Cross et al., 2011*; *Kapuy et al., 2009*; *Pomerening et al., 2003*), but proteins or other molecules that couple cell size to cell cycle activation (e.g. 'sizer' proteins) have been more difficult to identify (*Schmoller et al., 2015*).

No consensus exists for the mechanisms underlying cell size control in budding yeast, though several molecular pathways have been found to impact cell size. These include a regulatory loop involving accumulation of G1 cyclins and the removal of the inhibitory protein Whi5p from the transcriptional activator SBF that controls S phase entry (*Costanzo et al., 2004*; *de Bruin et al., 2004*; *Di Talia et al., 2009*; *Liu et al., 2015*; *Schmoller et al., 2015*). A phosphatase, Rts1p, appears to function upstream of this loop through the transcription factor Ace2 (*Di Talia et al., 2009*; *Zapata et al., 2014*). Budding yeast cell size control is also influenced by ribosome biogenesis that acts to modulate growth rate and the critical size at which cells enter S-phase (*Bernstein et al., 2007*; *Gomez-Herreros et al., 2013*; *Jorgensen et al., 2004*). It is still unclear what is the parameter monitored by size checkpoints in budding yeast. In fission yeast, it was proposed that cell size is controlled by polar localization of the kinase Pom1 that establishes an inhibitory gradient which blocks

**eLife digest** Most cells are programmed to maintain a certain size. This property, known as size control, is achieved by balancing growth and division, such that a cell will only divide after it reaches a certain size. However, and despite years of research, it is largely unknown how cells sense their size (or growth) to be able to divide accordingly. One theory proposes that there is a "sizer" protein inside cells, and that cells measure the abundance of this protein and use it to link cell size to the process of division. However, the existence of such a protein remained unproven.

Li, Liu et al. have now used the cells of the green alga Chlamydomonas to identify a candidate sizer protein. Chlamydomonas cells, like many other algae, can grow to become very large mother cells that then divide one or more times in succession to produce many daughter cells. Larger mother cells undergo more divisions than smaller mother cells in order to produce daughter cells of a correct size. Using a range of genetic and biochemical techniques, Li, Liu et al. identified a protein that is produced in Chlamydomonas cells just before they begin to divide. Larger mother cells contain more of this protein than smaller cells and the protein encourages cells to divide. For example, mutant cells that lack this protein divided too few times, while cells that produce too much of it divided too many times.

The protein, called CDKG1, belongs to a family of proteins that regulate cell division in many organisms. CDKG1 is a kinase – an enzyme that alters the activity of other proteins by adding a phosphate group on to them. In Chlamydomonas, CDKG1 couples cell size to cell division by altering the activity of an important protein called the retinoblastoma-related protein that controls cell division in numerous organisms. This protein is also frequently disrupted in cancers in humans.

These findings shed new light on a molecular pathway for size control. Future work will need to determine how the accumulation of CDKG1 links to the size of a mother cell and how it is inactivated once daughter cells reach the appropriate size.

the initiation of cytokinesis in the mid-cell region, thereby establishing a minimum length for cells prior to division (*Martin and Berthelot-Grosjean, 2009*; *Moseley et al., 2009*). However, this result has been challenged by new work showing that cells may instead sense their surface area through the local concentration of a kinase cdr2p (*Deng et al., 2014*; *Pan et al., 2014*). One complication in yeasts is that size thresholds for cell cycle transitions are modulated by cell growth rates, a property that adds an additional layer of complexity to size control networks in fungi (*Ferrezuelo et al., 2012*; *Turner et al., 2012*).

The unicellular green alga *Chlamydomonas reinhardtii* (Chlamydomonas) is a well-developed model organism (*Harris, 2001*) that is highly amenable to the investigation of cell-size control (*Umen, 2005*). Like many chlorophyte algae and diverse unicellular eukaryotes, Chlamydomonas cells proliferate using a multiple fission cell cycle (*Bisova and Zachleder, 2014*; *Cavalier-Smith, 1980*; *Cross and Umen, 2015*). Multiple fission is characterized by a prolonged G1 period during which cells can grow more than ten-fold in size. At the end of G1 mother cells undergo a series of rapid alternating S phases and mitoses (S/M) to produce $2^n$ uniform-sized daughters (*Umen, 2005*). Size control is evident during S/M because larger mother cells divide more times than smaller mother cells (*Craigie and Cavalier-Smith, 1982*; *Donnan and John, 1983*). Although size control mutants have been identified as described below, the mechanisms by which mother cells 'count' the correct number divisions or regulate daughter cell-size remain unclear. A second key attribute of multiple fission is that in diurnally-synchronized cultures growth occurs during the light period, while S/M phase occurs during the dark period with no additional growth of newborn daughter cells until the next light period. Under these conditions daughter cell-size is a direct readout of the mitotic size control mechanism (*Umen, 2005*).

Cell size control in Chlamydomonas also occurs during mid-G1 at a checkpoint termed *commitment*. Daughter cells pass commitment when they have attained a minimum size that is about twice their average birth size. Committed cells remain in G1 for an additional five to eight hours where they may continue to grow, and will proceed to divide at least once, even if no further growth takes

place (*Cross and Umen, 2015*; *Donnan and John, 1983*; *Oldenhof et al., 2007*; *Spudich, 1980*; *Umen, 2005*).

Previously we found that the Chlamydomonas retinoblastoma (RB) tumor suppressor pathway regulates both commitment size and cell division number (*Fang et al., 2006*; *Umen and Goodenough, 2001*). Mutants that are missing the retinoblastoma related (RBR) protein MAT3 pass commitment at an abnormally small size, remain in G1 for a normal post-commitment length of time, and then divide too many times during S/M to produce small daughters. These defects can be suppressed by mutations in *DP1* or *E2F1* that encode subunits of a conserved heterodimeric E2F-DP transcription factor that binds directly to MAT3/RBR to form a stable complex (*Fang et al., 2006*; *Olson et al., 2010*). To date no upstream regulators that integrate cell size information into the RBR pathway have been identified.

Here we describe CDKG1, a mitotic sizer protein that functions through the RBR pathway. CDKG1 is a nuclear-localized, D-cyclin dependent MAT3/RBR kinase whose mutant and mis-expression phenotypes indicate that its abundance is limiting for mother cell division number and mitotic size control. The production of CDKG1 was found to scale with mother cell size and was partially regulated through its long 3′ untranslated region. After each round of mitosis the amount of CDKG1 protein per nucleus decreased until it disappeared upon mitotic exit. Cell-size-dependent production of regulatory proteins is a potentially general means of linking cell size to downstream cell cycle events.

## Results

### CDKG1 is required for mitotic size control

In order to identify size regulators in Chlamydomonas we performed an insertional mutagenesis screen using the selectable marker *NIT1* to generate tagged mutants in a *nit1-305* background (*Tam and Lefebvre, 1995*). Direct screening of Nit+ insertion lines for size defects identified several mutants with large-cell phenotypes that were termed *lrg* mutants. Two independent allelic insertions, *lrg1-1* and *lrg1-2*, were mapped and found to disrupt the *CDKG1* gene (Cre06.g271100) (*Figure 1A,B*, and *Figure 1—figure supplement 1A*). CDKG1 was previously annotated as a Chlamydomonas-specific cyclin dependent kinase (*Bisova et al., 2005*), and for the remainder of this work we refer to the two insertion alleles as *cdkg1-1* and *cdkg1-2*.

Both *cdkg1* insertion alleles have associated chromosomal deletions, the smaller of which (*cdkg1-2*) removes most of *CDKG1* and part of an adjacent gene, *URH1* (Cre06.g271050), encoding a putative nucleoside hydrolase (*Mitterbauer et al., 2002*) (*Figure 1B* and *Figure 1—figure supplement 1A*). However, a genomic fragment containing only the full length *CDKG1* locus was able to complement the cell-size phenotype of *cdkg1-2* (*Figure 1C*) and restored *CDKG1* expression (*Figure 1D* and *Figure 1—figure supplement 1C,D*).

Large-cell phenotypes can result from either delayed cell cycle progression or from size checkpoint defects (*Mahjoub et al., 2002*; *Umen, 2005*). In order to distinguish these possibilities we examined daughter cells from dark-shifted *cdkg1-2* cultures and found them to be larger than wild-type daughters due to inadequate numbers of mitotic divisions (*Figure 1E*). We synchronized *cdkg1-2* and compared its cell cycle progression to similarly synchronized wild type cells (*Figure 1—figure supplement 1E*). The *cdkg1-2* cells grew comparably to wild type and passed Commitment in a size range slightly larger than wild type ($\sim$205–225 $\mu m^3$ versus $\sim$190–210 $\mu m^3$), though Commitment was attained earlier (2.5 hr vs. 4 hr) because *cdkg1-2* daughters are born larger than wild type daughter cells (*Table 1* and *Figure 1—figure supplement 1E*). We observed no obvious timing defects for *cdkg1* mutants for entry into and passage through S/M phase. The relatively normal Commitment size for *cdkg1-2* distinguishes it from *mat3-4*, *dp1-1* and *e2f1-4* that have altered Commitment sizes (*Fang et al., 2006*; *Umen and Goodenough, 2001*). After Commitment, *cdkg1* strains stayed in G1 phase for $\sim$8 hr indicating a normal delay period before starting S/M (*Figure 1—figure supplement 1E*). Together these results show that the large-cell phenotype of *cdkg1* mutants is due to defective size control during S/M and not caused by slow or impaired cell cycle progression.

In order to determine the relationship between the functions of *CDKG1* and *MAT3/RBR* we constructed a *cdkg1-2 mat3-4* double mutant strain. The double mutants were as small as *mat3* single mutants (*Figure 1F*), a relationship that is opposite to that seen in *dp1 mat3* double mutant strains

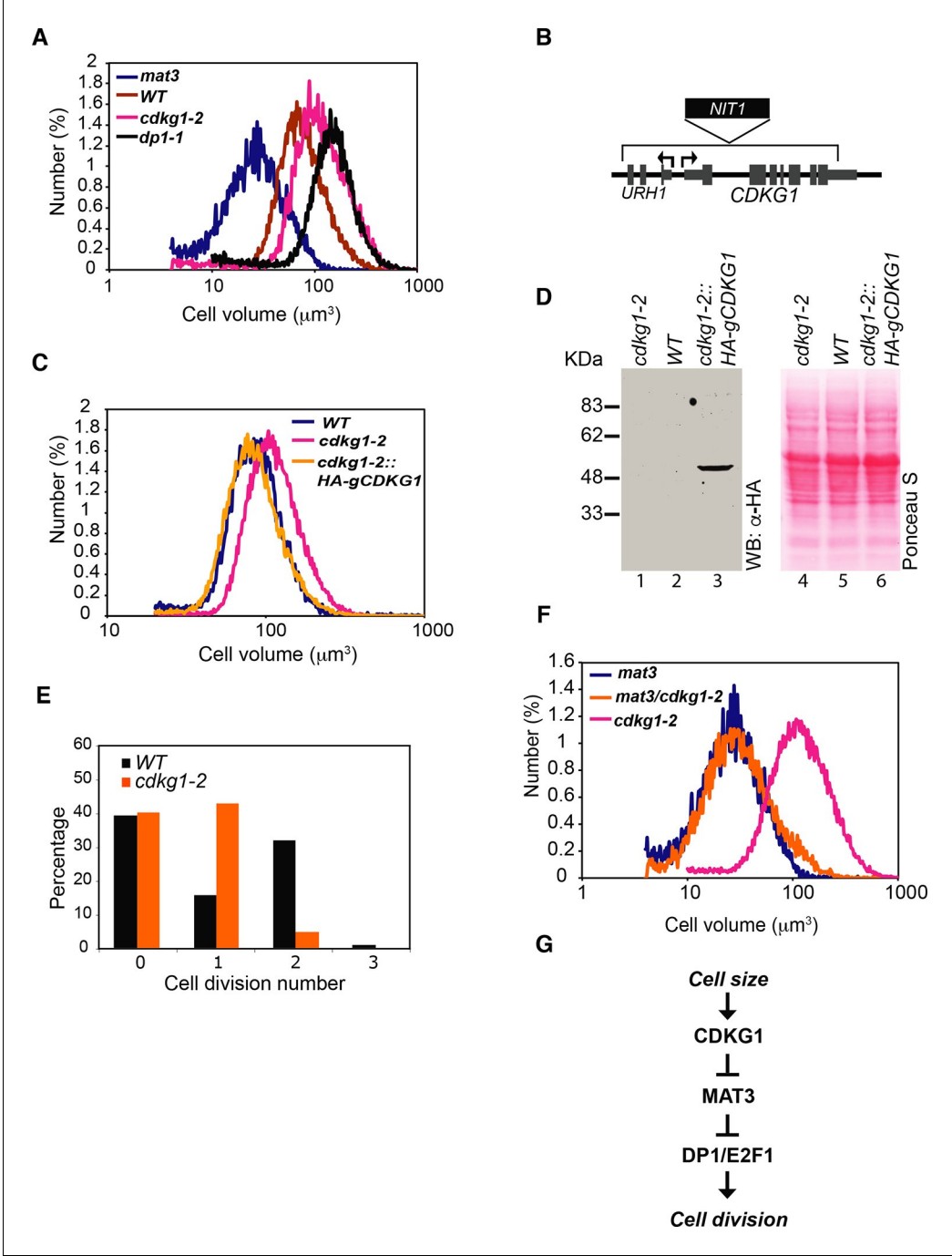

**Figure 1.** CDKG1 functions in size control upstream of MAT3/RBR. (**A**) Size distributions of daughter cells from indicated strains. (**B**) Schematic of *CDKG1* locus and *cdkg1-2* mutation that has a partial genomic deletion (in brackets) and insertion of the *NIT1* marker (shown in black). Tall gray bars, exons; narrow gray bars, untranslated regions; narrow black bars, introns and intergenic regions of *CDKG1* and *URH1*. Black arrows indicate transcription starts for *CDKG1* and *URH1*. (**C**) Size distributions of daughter cells from wild type (WT), *cdkg1-2* and complemented *cdkg1-2::HA-gCDKG1* strains. (**D**) Lanes 1–3, Western blot with anti-HA antibodies of total protein from *cdkg1-2* (lane 1), wild type (lane 2) and complemented *cdkg1-2::HA-CDKG1* (lane 3) strains using anti-HA antibodies. A single band at the predicted mass for HA-CDKG1 (~50kDa) is in Lane 3. Lanes 4–6, blot stained with Ponceau S to visualize total loaded protein. (**E**) Unsynchronized wild type and *cdkg1-2* cells were plated on HSM agar in the dark and division numbers were scored by percentage in each category: 0—no division (pre-Commitment), 1—one division (two daughters), 2—two divisions (four daughters), and 3—three divisions (eight *Figure 1 continued on next page*

*Figure 1 continued*

daughters). (F) Size distributions of daughter cells from *mat3-4*, *mat3-4 cdkg1-2*, and *cdkg1-2* strains. (G) Epistasis hierarchy of size control mutants in Chlamydomonas.

The following figure supplement is available for figure 1:

**Figure supplement 1.** Cell cycle progression and complementation of *cdkg1-2*.

that have large cells (*Fang et al., 2006*). This epistasis relationship indicates that CDKG1 acts genetically upstream of the MAT3/RBR complex to control cell size (*Figure 1G*).

## CDKG1 is a novel cyclin dependent kinase

The predicted CDKG1 protein has a unique N-terminal domain of ~90 amino acids followed by a kinase domain with conserved catalytic motifs and homology to the CDK family (*Figure 2A*, and *Figure 2—figure supplement 1*) (*Bisova et al., 2005*), but the canonical PSTAIRE motif of CDK1 that constitutes part of its cyclin binding interface (*Morgan, 1997*) is replaced by SDSTIRE in CDKG1. Previous phylogenetic analyses suggested that CDKG1 is a member of a unique CDK sub-family (*Bisova et al., 2005*). To better understand its phylogenetic relationship to other CDKs, we generated a maximum-likelihood phylogeny of kinase domains from CDKG1 and other CDKs including metazoan RB kinases (*Figure 2B*) (*Bisova et al., 2005*). Our results show that CDKG1 and its distant paralog, CDKG2, belong to a novel subfamily of CDKs that are distinct from the eukaryotic CDK1/CDKA clade and more divergent from CDK1/CDKA than the plant-specific CDKB and animal-specific CDK2 subfamilies. On the other hand, the CDKG family is more closely related to core cell cycle CDKs of animals and plants than the CDK4/6 family of RB kinases from animals. Together these data indicate that CDKG1 might function as a specialized cell cycle CDK.

To determine whether CDKG1 could fulfill the role of a cell cycle CDK, we expressed its cDNA in a budding yeast *cdc28-13* strain that has a temperature sensitive lethal mutation in its CDK1 protein (Cdc28p) (*Reed and Wittenberg, 1990*). Wild-type *CDKG1* could complement *cdc28-13* at 37°C, but mutations in *CDKG1* that were predicted to make it catalytically dead (K125R) or non-activatable (T254A) (*Morgan, 1997*) did not rescue the temperature-sensitive phenotype (*Figure 2C* and *Figure 2—figure supplement 1*). These results indicate that CDKG1 retains essential regulatory features of cell cycle CDKs.

## CDKG1 binds to D-type cyclins and phosphorylates the RB homolog MAT3

To identify its potential cyclin partners we tested members of each of the major classes of Chlamydomonas cyclins for interaction with CDKG1 in a yeast two-hybrid (Y2H) assay (*Miller and Stagljar, 2004*). Chlamydomonas has one A-type cyclin (CYCA1), one B-type cyclin (CYCB1), one hybrid-A/B-type cyclin (CYCAB1), and four D-type cyclins (CYCD1-CYCD4) (*Bisova et al., 2005*; *Prochnik et al., 2010*). Full-length cDNAs of *CYCA1, CYCB1, CYCD2, CYCD3 and CYCD4* were successfully amplified through RT-PCR and used for Y2H testing with CDKG1 (*Figure 3A* and *Figure 3—figure*

**Table 1.** Daughter cell size, commitment size and growth rate for indicated strains

| Strain | Daughter cell size ($\mu m^3$) | Commitment size ($\mu m^3$) | Doubling time (hr) |
|---|---|---|---|
| wild type | 75 ± 3 | 195 ± 14 | 5.89 ± 0.7 |
| *cdkg1-2* | 95 ± 6 | 215 ± 10 | 5.62 ± 0.7 |
| *cdkg1-2::HA-gCDKG1* | 73 ± 7 | 190 ± 12 | nd |
| *PSAD::CDKG1* | 56 ± 2 | 185 ± 3 | nd |
| *dp1-1*[a] | 104 ± 5 | 237 ± 4 | nd |
| *mat3-4*[b] | 25 ± 2 | 110 | nd |

± : Standard deviation; nd: not determined; a from (*Fang et al., 2006*); b from (*Umen and Goodenough, 2001*)

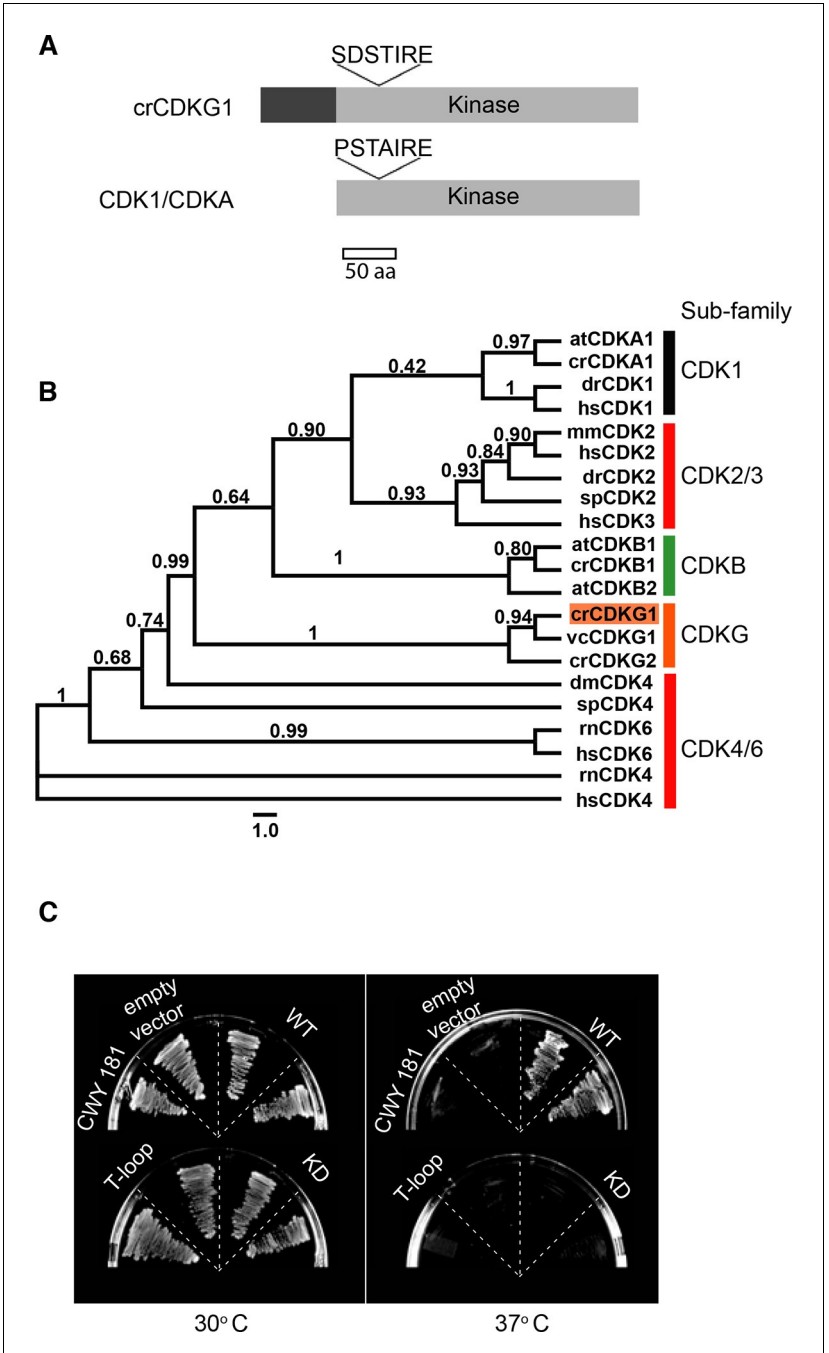

**Figure 2.** *CDKG1* encodes a novel cyclin dependent kinase. (**A**) Schematic representation of Chlamydomonas CDKG1 and CDK1/CDKA structures. The unique N-terminus (dark grey bar) and relative location of variant SDSTIRE motif of CDKG1 and PSTAIRE motif from CDK1/CDKA are shown. Light grey bar represents conserved kinase domain. (**B**) Unrooted Maximum likelihood phylogenetic tree of CDK domain alignments. Sub-families are indicated by color-coded vertical bars and labeled. Species abbreviations are: at, *A. thaliana;* cr, *C. reinhardtii;* dm, *D. melanogaster;* dr, *D. rerio;* hs, *H. sapiens;* mm, *M. musculus;* sp, *S. purpuratus;* rn, *R. norvegicus;* vc, *V. carteri.* Approximate likelihood ratio support is indicated for each node. (**C**) Growth of yeast strain CWY181 *cdc28-13* and derivatives that were transformed with empty vector only, wild type *CDKG1* (WT), and versions of *CDKG1* containing predicted kinase inactive or T-loop mutations and grown at the permissive (30°C, left panel) or restrictive (37°C, right panel) temperatures.

The following figure supplement is available for figure 2:

*Figure 2 continued on next page*

*Figure 2 continued*

**Figure supplement 1.** CDKG1 multiple sequence alignment and sequence features Multiple sequence alignment of *Chlamydomonas* CDKG1 (crCDKG1), CDKA1 (crCDKA1) and human CDK1 (hsCDK1).

*supplement 1A*). *CYCA1, CYCB1, CYCD2* and *CYCD3* are most highly expressed cyclins in Chlamydomonas while *CYCD4* and *CYCD1* are expressed at much lower levels (*Bisova et al., 2005*; *Zones et al., 2015*). All three D-cyclins that could be tested by Y2H interacted with CDKG1, but CYCD2 and CYCD3 had much stronger signals than CYCD4. The putative S-phase and mitotic cyclins CYCA1 or CYCB1 had no detectable interactions with CDKG1 in this assay, but both were able to interact with CDKA1 and CDKB1 as anticipated from previous studies (*De Veylder et al., 2007*).

The D-type cyclins in Chlamydomonas contain a conserved LxCxE motif in their N-termini (*Bisova et al., 2005*) that is known to be important for binding RBR proteins (*Lee et al., 1998*). We tested whether CYCD2, CYCD3 or CDKG1 could interact with MAT3/RBR in a Y2H assay (*Miller and Stagljar, 2004*). All three proteins tested positive for binding to MAT3/RBR by Y2H, though the D-cyclins interacted with MAT3/RBR only weakly (*Table 2*). Substituting AxAxA for LxCxE in CYCD3 disrupted its interaction with MAT3/RBR, suggesting that this motif is important for binding (*Figure 3—figure supplement 1C*).

Specific binding of CYCD2 and CYCD3 to CDKG1 were confirmed using a quantitative immunoprecipitation (IP) assay with *in vitro* translated (IVT) proteins (*Figure 3B*). Interactions between CDKG1, CYCD2, CYCD3 and MAT3/RBR were tested further using an *in vitro* pull-down assay. We found that IVT CDKG1 bound to recombinant GST-MAT3/RBR and that it did not require its unique N-terminal extension or a D-cyclin partner to do so (*Figure 3C*, lanes 1–4). CYCD3 was able to bind MAT3/RBR directly (*Figure 3C*, lanes 5–6). However, CYCD2 only bound to MAT3/RBR in the presence of CDKG1 and perhaps did so indirectly (*Figure 3C*, lanes 7–9).

The protein interactions we identified suggest that CDKG1 might directly interact with and phosphorylate MAT3/RBR as its substrate. To test this possibility, we first developed an *in vitro* kinase assay with partially purified IVT HA-CDKG1 or a predicted kinase dead allele (HA-CDKG1$^{kd}$), with and without added Myc-tagged CYCD3 (myc-CYCD3). GST-MAT3, GST, and histone H1 were used as substrates. By itself HA-CDKG1 exhibited some GST-MAT3 kinase activity that was consistently stimulated by around 1.7-fold in the presence of CYCD3 (*Figure 3D*, lanes 2 and 4, *Figure 3—figure supplement 1D*). The phosphorylation of GST-MAT3 was not observed in reactions with HA-CDKG1$^{kd}$ and was therefore dependent on having an active CDKG1 kinase domain (*Figure 3D*, lanes 1,3 versus 2,4). HA-CYCD2 did not stimulate HA-CDKG1 kinase activity above background levels (data not shown), perhaps owing to weaker binding to MAT3/RBR (*Figure 3C*). *CYCD3* is the most abundant D-type cyclin mRNA in Chlamydomonas (*Figure 3—figure supplement 1A*) and is expressed in a pattern similar to that of *CDKG1* mRNA, making CYCD3 a likely partner for CDKG1 during S/M.

We also developed a kinase assay using immunoprecipitated CDKG1 from complemented *cdkg1-2::HA-gCDKG1* cultures or from a wild-type strain expressing a *HA-CDKG1$^{kd}$* transgene to test the activity of native CDKG1 complexes against MAT3. Immunoprecipitated protein was incubated, with recombinant GST-MAT3, GST only, or histone H1 as substrates. Histone H1 and GST-MAT3, but not GST, were phosphorylated by kinase activity present in the HA-CDKG1 IP pellet (*Figure 3F*). In contrast, barely detectable kinase activity was observed from equivalent amounts of HA-CDKG1$^{kd}$ protein (*Figure 3F*).

To determine whether CDKG1 interacts with MAT3/RBR *in vivo* we used a co-immunoprecipitation (Co-IP) assay. HA-tagged MAT3/RBR (3XHA-MAT3) from a complemented *mat3-4::HA-MAT3* strain (*Olson et al., 2010*) was subject to IP, and the IP pellets were probed on Western blots with polyclonal antibodies raised against recombinant CDKG1. CDKG1 was found associated with MAT3/RBR only in S/M phase cells (*Figure 3E*) where it is most highly expressed (see below) and not in a control extract prepared from untagged cells in S/M phase (*Figure 3—figure supplement 1E*). Previously we were able to detect cell-cycle controlled phosphorylation of MAT3/RBR that could be qualitatively observed by migration changes on SDS-PAGE gels in synchronized cultures (*Olson et al., 2010*). We constructed isogenic strains where a HA-tagged MAT3/RBR transgene was in a wild type or *cdkg1-2* background and compared the phosphorylation levels in daughters and in

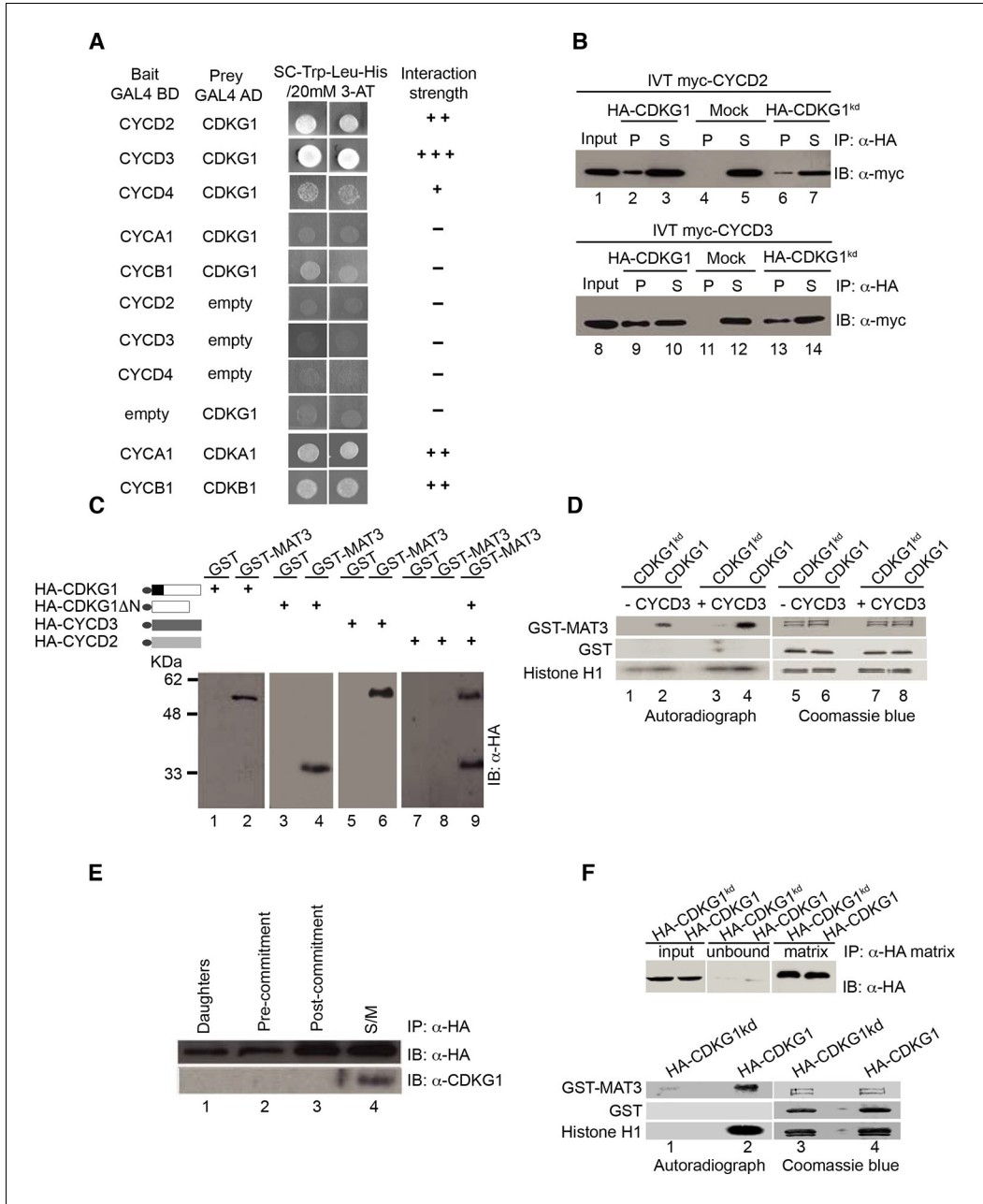

**Figure 3.** CDKG1 interacts with D cyclins and phosphorylates the Chlamydomonas RBR protein MAT3. (**A**) Yeast two hybrid (Y2H) assay with Gal4 activation domain (AD) fusions (prey) and Gal4-DNA binding domain (DB) fusions (bait) indicated in the first two columns. Empty indicates vector only. Growth of two independent co-transformants is shown, and the relative strength of interaction is indicated by -, no interaction, +, weak interaction, ++, strong interaction, and +++, very strong interaction. (**B**) Co-immunoprecipitation (Co-IP) of *in vitro* translated (IVT) myc-tagged CYCD2 (upper panel) or CYCD3 (lower panel) incubated with HA-tagged CDKG1, anti-HA beads only (mock), or HA-CDKG1kd (kinase dead mutation). Western blots were done using anti-myc antibodies with input samples (lanes 1,8), IP pellets (P, lanes 2, 4, 6, 9, 11, 13) and IP supernatants (S, lanes 3, 5, 7, 10, 12, 14). (**C**) Anti-HA Western blot of *in vitro* translated (IVT) proteins bound to GST or GST-MAT3. HA-CDKG1 (HA-G1, lanes 1, 2), HA-CDKG1 kinase domain only (HA-G1ΔN, lanes 3, 4), HA-CYCD3 (HA-D3, lanes 5, 6), HA-CYCD2 (HA-D2, lanes 7, 8) and HA-G1ΔN+HA-CYCD2 (lane 9). Schematic structures of IVT *proteins* are shown on the left with the HA tag indicated by an ellipse. (**D**) *In vitro* kinase assay using IVT full length HA-CDKG1 (lanes 2, 4, 6, 8) or a predicted kinase dead version (HA-CDKG1kd, lanes 1, 3, 5, 7) co-immunoprecipitated with (+CYCD3) or without (–CYCD3) IVT CYCD3. Left panel, autoradiograph of proteins separated by SDS-PAGE after the kinase assay (lanes 1–4); right panel, Coomassie blue stained bands corresponding to the autoradiograph (lanes 5–8). Substrate proteins used

*Figure 3 continued on next page*

*Figure 3 continued*

are labeled on the left. (E) Western blots of anti-HA IP pellets from Chlamydomonas strain *mat3-4* complemented with a *HA-MAT3* construct (*Olson et al., 2010*) and synchronized in a 14 hr light/10 hr dark cycle. Anti-HA (top panel) and anti-CDKG1 (bottom panel) blots prepared with extracts from daughter cells (0 hr light), pre-Commitment cells (4 hr light), post-Commitment cells (10 hr light) and S/M phase cells (1 hr dark). (F) Kinase assay using immunoprecipitated HA-CDKG1 or HA-CDKG1$^{kd}$ from Chlamydomonas strain *cdkg1-2::HA-gCDKG1* or *HA-CDKG1$^{kd}$*. Upper panel, Western blots of total input and IP products. Lower panel, autoradiograph of proteins separated with SDS-PAGE (lane 1 and 2) and coomassie blue stained bands corresponding to the autoradiograph (lane 3 and 4).

The following figure supplements are available for figure 3:

**Figure supplement 1.** Expression profiles and interactions between D cyclins, CDKG1 and MAT3/RBR during the cell cycle.

**Figure supplement 2.** HA-MAT3 phosphorylation in *cdkg1-2* cells Total protein extracts from indicated strains at daughter (D) or S/M stage (M) were separated with SDS-PAGE or phos-tag SDS PAGE gels, followed by western blotting with anti-HA to detect HA-MAT3.

---

S/M phase populations. Although we observed S/M phase hyperphosphorylation of MAT3/RBR, we did not observe a detectable change in MAT3/RBR phosphorylation in the *cdkg1-2* mutant compared with controls. These results indicate that CDKG1 is not the predominant kinase for MAT3/RBR, and that like its animal homologs, MAT3/RBR is likely to be a substrate for multiple CDKs (*Figure 3—figure supplement 2*). Taken together our data suggest that MAT3/RBR is a substrate for CDKG1 during S/M phase but that CDKG1 is not the dominant kinase for the MAT3/RBR complex.

## CDKG1 mRNA and protein accumulation are cell-cycle regulated

The expression pattern of CDKG1 during the cell cycle was determined by synchronizing *cdkg1-2::HA-gCDKG1* cultures and Western blotting of samples prepared from different time points. To facilitate comparisons, lysates were loaded on two blots, one with equal amounts of total protein in each lane (*Figure 4A*, upper panels) and one with equal numbers of cells in each lane (*Figure 4A*, lower panels). Loading by equal protein normalizes samples and provides a comparison of overall cellular protein concentration at each time point. However, because cells increase in mass by about ten-fold during G1 phase, loading by equal cell number provides an indication of how much total CDKG1 protein is present in each cell. HA-CDKG1 was nearly undetectable during early G1 prior to Commitment (*Figure 4A* lanes 1–4) and was expressed at very low levels in post-Commitment G1 cells (*Figure 4A* lanes 5–7). Peak expression of HA-CDKG1 occurred just before the beginning of S/M. Its level then decreased rapidly in post-mitotic cells suggesting rapid turnover upon entry into G0/G1 phase.

**Table 2.** Summary of yeast two hybrid results for CDKG1, D-cyclins and MAT3

| Bait \ Prey | CDKG1 | CYCD2 | CYCD3 | CYCD3$^{AxAxA}$ | CYCD4 | MAT3 | Empty |
|---|---|---|---|---|---|---|---|
| CDKG1 | NA | ++ | +++ | +++ | + | + | - |
| CYCD2 | ++ | NA | NA | NA | NA | + | - |
| CYCD3 | +++ | NA | NA | NA | NA | + | - |
| MAT3 | + | + | + | - | NA | NA | - |
| Empty | - | - | - | - | - | - | NA |

+++: Strong interaction

++: Medium interaction

+: Weak interaction

-: No interaction

NA: Not available

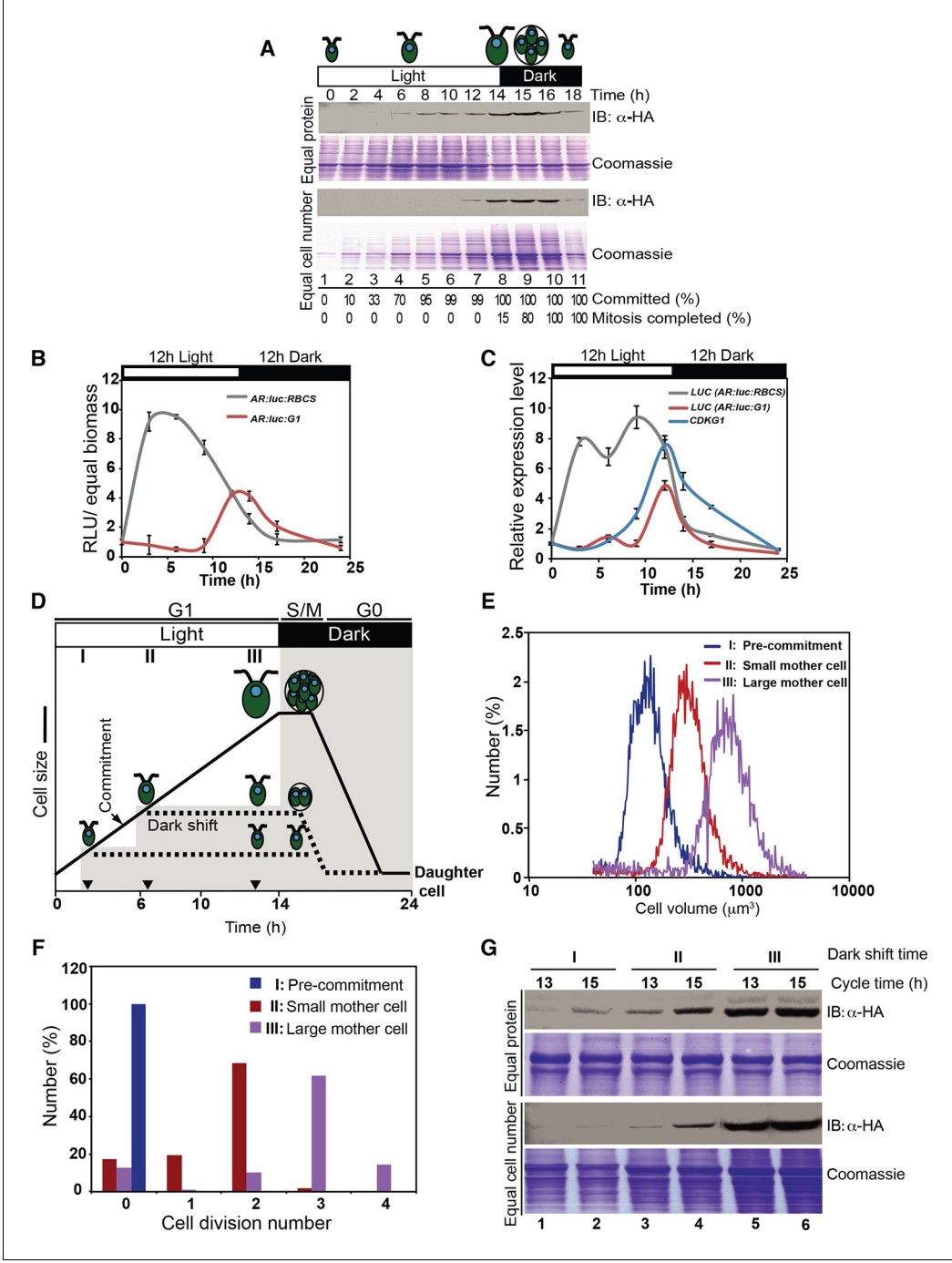

**Figure 4.** CDKG1 protein accumulation is cell-cycle and cell-size regulated. (**A**) Anti-HA Western blot of total protein extracts from synchronized *cdkg1-2::HA-gCDKG1* grown under a 14 hr light/10 hr dark regime. The open bar represents the light period and the black bar represents the dark period. Samples for immunoblotting were collected at indicated time above each lane. Upper two panels, Western blot and Coomassie staining with equal amount of total protein per lane; lower two panels, Western blot and Coomassie staining with equal cell number per lane. The fractions of cells that were past commitment and had completed mitosis at each time point are shown under each lane at the bottom. (**B**) Luciferase activity (RLU) was normalized to biomass and plotted from synchronous wild type cells expressing *AR:luc:RBCS* either or *AR:luc:G1*. (**C**) Quantitative RT-PCR of mRNAs from strains in panel (**B**) and ofendogenous *CDKG1*. Expression is normalized to 1 at the first time point. (**D**) Schematic representation of the method to generate different-sized mother cells. A synchronized *cdkg1-2::HA-gCDKG1* culture was entrained to a 14 hr light/10 hr dark regime. Cells passed commitment at ~5 hr as marked. At three
*Figure 4 continued on next page*

*Figure 4 continued*

hours in the light the culture was split into three parts, the first of which was dark shifted immediately to produce a population of pre-Commitment cells (I), the second of which was dark-shifted at 7 hr to produce a population of small mother cells (II), and the third of which stayed in the light for 14 hr to produce a population of large mother cells (III). The black curve and dashed lines show cell sizes in each of the three samples. Black inverted triangles mark time points for dark-shift of sub-cultures I and II. (E) Cell size distributions of populations described in panel (D) taken at 13 hr. (F) Bar graph of cell division numbers from populations described in panel (D). (G) Anti-HA Western blot and Coomassie staining of total protein extracts from populations I, II and III at 13 hr (pre-mitotic, lanes 3 and 5) and 15 hr (mitotic, lanes 4 and 6). Samples from culture (I) were used as a non-dividing control (lanes 1 and 2). Upper and lower panels were loaded as described for panel (A).

The following figure supplements are available for figure 4:

**Figure supplement 1.** Control of CDKG1 mRNA abundance by mother cell size Bar graph shows *CDKG1* mRNA levels from synchronized cultures covering the time period of S/M (13 hr to 15 hr), plus a post-mitotic sample (18 hr).

**Figure supplement 2.** Quantitation of HA-CDKG1, histone H3 and alpha-tubulin from mother cells of different sizes.

Quantitative reverse transcription and PCR (qRT-PCR) performed on a sample time course similar to that in *Figure 4A* showed that *CDKG1* mRNA levels during the cell cycle approximately match its protein accumulation pattern (*Figure 1—figure supplement 1D*). The *CDKG1* 3' UTR is unusually long (1.7 kb) and uridine-rich (27% U) compared to the average for Chlamydomonas genes (595 bp for 3' UTR, 19% for U content) (*Shen et al., 2008*) (*Figure 1—figure supplement 1A*) suggesting that its 3'UTR might be involved in controlling mRNA or protein abundance. To investigate the regulation of *CDKG1* expression we generated reporter constructs in which a *Gaussia princeps* luciferase gene (*Gluc*) (*Ruecker et al., 2008*) was expressed under the control of promoter and 3' UTR combinations derived from *CDKG1* (*pG1:luc:G1*) or from a construct containing the *HSP70A-RBCS2* (AR) hybrid promoter and *RBCS2 3'UTR* (*AR:luc:RBCS*) (*Ruecker et al., 2008*; *Schroda et al., 2000*). Reciprocal swaps of promoter and 3'UTR sequences generated *AR:luc:G1* and *pG1:luc:RBCS*. All four constructs were transformed into wild-type cells and random transformants were tested to identify positive clones. When the *CDKG1* promoter (pG1) was used to express luciferase we were unable to detect expression above background even after screening hundreds of clones suggesting that pG1 is a relatively weak promoter. However, for both *AR:luc:RBCS* and *AR:luc:G1* we identified transformants with luciferase activity. Using these strains we measured luciferase activity in synchronized cultures growing in a 12 hr light/12 hr dark regime. *AR:luc:RBCS* transformants showed an immediate increases in luciferase activity at the beginning of the light period and slow decline that continued into the dark period. In contrast, *AR:luc:G1* expression remained low until the end of the light phase when it increased and peaked near the time of cell division (*Figure 4B*). We measured mRNA from both of these luciferase constructs using qRT-PCR and compared them to the mRNA levels of the endogenous *CDKG1* mRNA in synchronized cultures. *AR:luc:G1* and endogenous *CDKG1* mRNA showed the same accumulation profiles suggesting that the *CDKG1* 3' UTR sequence exerts its effects on mRNA accumulation by regulating the stability of mRNAs in a cell cycle dependent manner (*Figure 4C*).

## CDKG1 production scales with mother cell-size

We next determined how the abundance of CDKG1 is related to mother cell size. To do so, we synchronized the *cdkg1-2::HA-gCDKG1* strain and generated two populations of mother cells— large and small (*Figure 4D and E*). The large mother cells (population III) were obtained by leaving the cultures in the light for a normal diurnal period of 14 hr, while the small mother cells (population II) were obtained by dark shifting a culture at 7 hr in the light. Both populations II and III divided at the same time (13–15 hr), but the large mother cells divided three or four times while the small mother cells divided one or two times (*Figure 4F*). A control population of cells (I) was dark-shifted prior to Commitment (3 hr light), and had no dividing cells (*Figure 4D and F*).

We prepared extracts from each of the three populations of cells sampled at 13 hr and 15 hr that correspond to late G1 and S/M respectively for populations II and III. HA-CDKG1 was barely detectable in non-dividing control cells from population I (pre-Commitment cells, *Figure 4G* lanes 1, 2) indicating that its expression is correlated with cell division. In contrast, both small and large mother cells expressed CDKG1 just prior to and during cell division (*Figure 4G* lanes 3–6). Moreover, the amount of HA-CDKG1 that accumulated in small mother cells (*Figure 4G*, lanes 3, 4) was lower than that in large mother cells (*Figure 4G* lanes 5, 6). This size-dependent difference was evident even when equal protein was loaded from each sample meaning that the total concentration of CDKG1 is higher in large versus small mother cells (*Figure 4G*, upper panels, lanes 3, 4 versus 5, 6). Thus both cellular concentration and total amounts of CDKG1 per cell scale allometrically (i.e. non-linearly) with mother cell size. This allometric scaling relationship was also reflected in *CDKG1* mRNA abundance that also increased relative to an internal control in large versus small mother cells (*Figure 4—figure supplement 1*). Compared to bulk protein observable on coomassie-stained gels CDKG1 accumulation does not scale linearly with mother cell size (*Figure 4G*). We also asked whether the CDKG1 accumulation pattern was distinct from two other proteins, histone H3 and alpha-tubulin, neither of which is expected to play a role in cell-size control. The alpha tubulin mRNA has a cell cycle regulated accumulation pattern with a peak during S/M (*Zones et al., 2015*) as does histone H3 (*Waterborg et al., 1995*). Quantitative Western blotting from the same populations of non-dividing cells, small mother cells and large mother cells revealed that the scaling relationship of CDKG1 with cell cycle stage and cell size is different from histone H3 and tubulin where the former protein had an approximately fixed amount per cell and the latter was produced proportionally to total protein biomass (*Figure 4—figure supplement 2*). Thus, the allometric accumulation pattern of CDKG1 where the amount made is not linear with respect to bulk protein and cell size suggests that its abundance in mother cells might be related to its role in cell size control.

## CDKG1 is nuclear-localized and is diluted and degraded during cell division

To gain more detailed insight into how CDKG1 is temporally regulated we determined its accumulation pattern in individual *cdkg1-2::HA-gCDKG1* cells using indirect immunofluorescence (IF) with anti-HA antibodies (*Figure 5A* and *Figure 5—figure supplement 1A*). Consistent with its expression pattern determined by Western blotting, HA-CDKG1 IF signal was only detected in late G1 and S/M cells (*Figure 5A* and *Figure 5—figure supplement 1B*). Nearly all detectable HA-CDKG1 staining was nuclear, though we did observe diffuse cytoplasmic staining in late G1 cells (*Figure 5—figure supplement 1C*).

We next followed the abundance of nuclear HA-CDKG1 using quantitative IF during the course of S/M in a population of large mother cells that underwent at least three rounds of cell division to produce eight daughters. We determined the relative nuclear concentration of HA-CDKG1 in dividing cells and found that it gradually decreased with each round of cell division (*Figure 5D*). To determine whether the decline of nuclear CDKG1 concentration was attributable to dilution and/or protein turnover we first measured the nuclear volumes (N) and total cell volume (C) from mitotic cells in which a nuclear-localized ble-GFP fusion protein (*Fuhrmann et al., 1999*) was constitutively expressed and used to mark the nucleus (*Figure 5B*). When we measured N/C in mitotic cells we found that it remained constant and that nuclear volume halved with each round of division (*Figure 5C*). This finding indicates that total nuclear volume in dividing cells is fixed, and that changes in CDKG1 concentration were due to a small but significant amount of net degradation as cells progressed through each round of division (*Figure 5A*). Importantly, all the CDKG1 protein that was synthesized in cells appears to have been made in mother cells just prior to S/M and it was not replaced as cells progressed through division. While the absolute concentration drop in CDKG1 was relatively shallow with each round of S/M, its concentration with respect to nuclear DNA dropped by more than two fold with each cell division (*Figure 5E*) and this ratiometric decrease could potentially be used as a threshold signal to control mitotic exit.

## Mis-expression of CDKG1 causes a small-cell phenotype

The large-cell phenotype of *cdkg1* mutants, the cell-size dependent scaling of pre-mitotic CDKG1 production, and the dynamic changes in CDKG1 nuclear abundance with each round of division

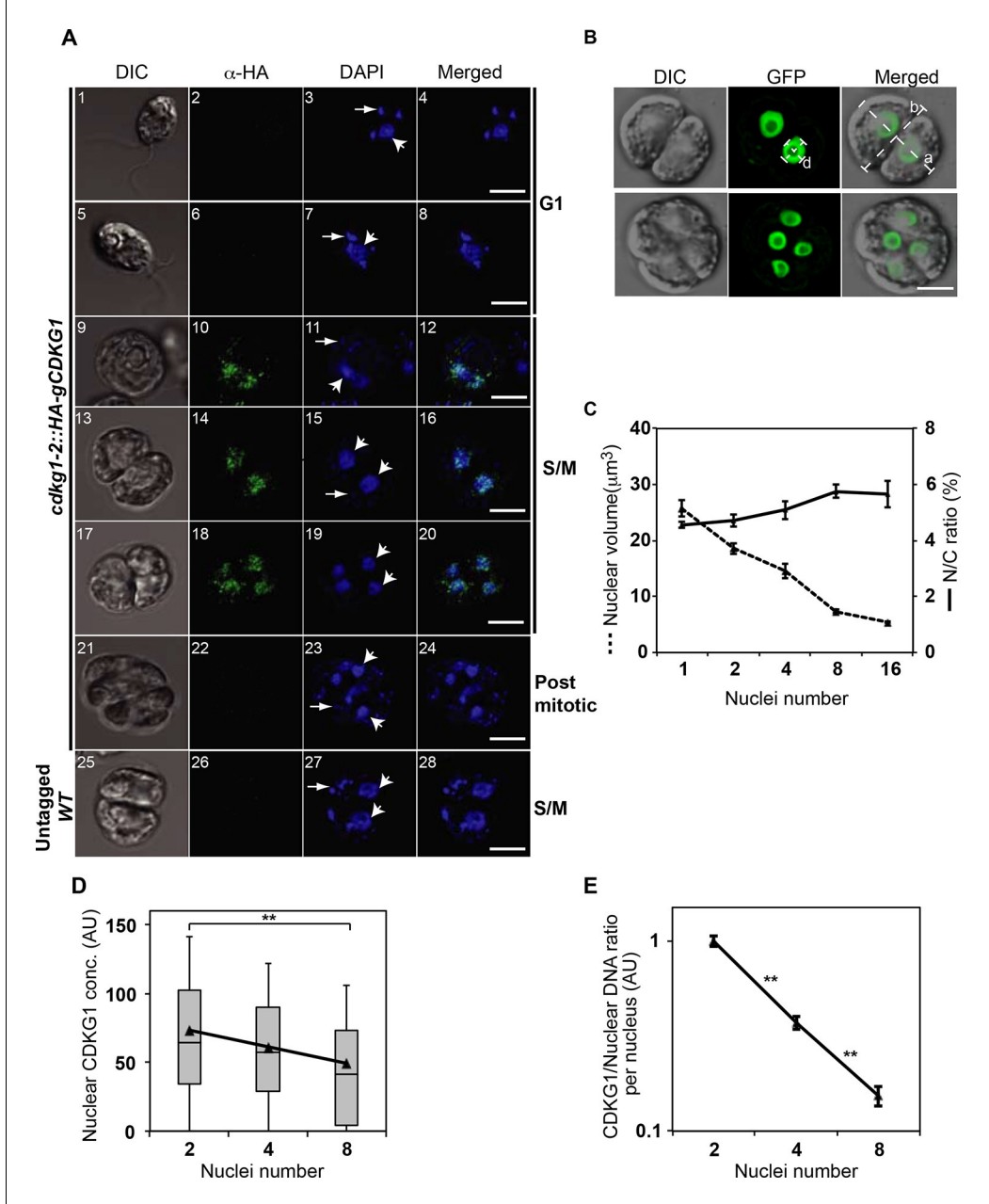

**Figure 5.** CDKG1 is nuclear localized and degraded as cells divide. (**A**) DIC and confocal immunofluorescence microscopy images of *cdkg1-2::HA-gCDKG1* (panels 1–24) and untagged wild type (WT) cells (panels 25–28). Synchronized cultures were fixed and immunostained for the HA epitope (green) and stained with DAPI (blue). Panels 1–8 are G1 phase cells, 9–20 are S/M phase cells, and 21–24 are post mitotic cells. Panels 25–28 are S/M phase cells from an untagged control strain. Wide arrows mark representative nuclei; thin arrows mark representative chloroplast nucleoids; Scale bar = 5 µm. Also see *Figure 5—figure supplement 1*. (**B**) Live synchronized S/M phase ble-GFP expressing cells were imaged by DIC or with a maximum-projection of Z stacks from the GFP channel. The diameter of a representative nucleus (d) and major (a) or minor (b) axes of the mother cell body are marked as white dashed lines. Scale bar = 6 µm. (**C**) Graph of nuclear volume and mean N/C ratio per daughter cell at different division stages plotted against nuclear number. Data are from at least 24 cells per group, except for the 16-cell stage (n=11). Error bars: S.E.M. No significant differences on N/C ratio were detected among groups using a one-way ANOVA test (p=0.09879, d=4). (**D**) Box-whisker plots of nuclear concentrations of HA-CDKG1 per nucleus against nuclei number (X-axis) from dividing cells at different cycle numbers. Black triangles represent mean values of nuclear HA-CDKG1 concentration per nucleus that used for generating the linear regression ($R^2$=0.99959). Data were collected from at least 68 cells per division stage. Error bar: S.D. **: two-

*Figure 5 continued on next page*

*Figure 5 continued*

tailed non-parametric t-test between cell groups with 2 and 8 nuclei (p<0.005). (**E**) Graph of (HA-CDKG1/nuclear DNA) from different division cycle numbers as described in (**D**). The ratio was calculated as total nuclear HA-CDKG1 immunofluorescence intensity per mitotic cell divided by the genome copy number within that cell cluster. The HA-CDKG1/nuclear DNA ratio in first group was set as 1 arbitrary unit (AU) and plotted against nuclear number. Error bar: S.E.M.. p values of non-parametric t test between adjacent samples are marked. **\*\*:** significant difference (p<0.0001).

The following figure supplement is available for figure 5:

**Figure supplement 1.** HA-CDKG1 immunolocalization.

suggests that this protein might be limiting for normal mitotic cell division number and size control. To test this idea, the coding region of the *CDKG1* cDNA was put under the control of a strong constitutive *PSAD* promoter and untranslated region (*Fischer and Rochaix, 2001*) to generate *PSAD: CDKG1*. *PSAD:CDKG1* was introduced into wild-type cells and several independent transformants expressing the cDNA were isolated. The *PSAD:CDKG1*-expressing lines had a small-cell phenotype (*Figure 6A* left panel) that was never seen in control lines transformed with an empty vector or in transformants that expressed a K125R kinase-dead version of *CDKG1* (*CKDG1^kd^*) (*Figure 6A* right panel, *Figure 2—figure supplement 1*). Consistent with CDKG1 influencing only the S/M phase cell-size control point, *PSAD:CDKG1*-expressing cells passed Commitment at close to wild-type size, ~185 μm³, but then underwent super-numerous divisions during S/M to produce small daughters (*Figure 6—figure supplement 1* and *Table 1*).

The relatively low frequency with which we recovered transformants expressing *PSAD:CDKG1* (<0.5%) led us to test an alternative vector system and at the same time include an epitope tag to allow detection of CDKG1 protein. We constructed a new vector, *PL23:HA-CDKG1,* that contains a HA tagged *CDKG1* cDNA whose expression is driven by the promoter and untranslated regions of a cytoplasmic ribosomal protein gene *L23* (López-Paz et al., 2016, in preparation). *PL23:HA-CDKG1* was transformed into wild-type cells and transformants with small daughters were identified at an increased rate over the *PSAD* vector (1–2%), all of which had detectable expression of HA-CDKG1 (*Figure 6B and C*). We next compared the HA-CDKG1 expression patterns from a complemented strain (*cdkg1-2::HA-gCDKG1*) where CDKG1 expression is controlled by its native promoters and UTRs, versus a wild-type strain expressing native CDKG1 and an exogenously integrated *PL23:HA-CDKG1* construct. In the complemented control strain *CDKG1* mRNA and HA-CDKG1 protein were detected almost exclusively in S/M phase cells with little or no expression in daughter cells (*Figure 6D and E*). In contrast, *PL23:HA-CDKG1*-expressing cells showed a detectable accumulation of *CDKG1* mRNA and HA-CDKG1 protein in both S/M phase cells and in daughter cells where it is never seen in controls (*Figure 6D and E*). As we observed for *PSAD:CKDG1^kd^*, expression of *PL23: HA-CDKG1^kd^* caused no cell size phenotype (*Figure 6—figure supplement 2C and D*). Mis-expression of CDKG1 is likely to be detrimental or unstable since the small-cell phenotype and expression of the transgene were both lost upon repeated passage of *PL23:HA-CDKG1* cultures (*Figure 6—figure supplement 2A and B*), and this property may also account for the overall low frequency of obtaining transformants that mis-express *CDKG1*. Taken together our data indicate that CDKG1 is a limiting component of the normal S/M size control mechanism in Chlamydomonas, that its abundance is tightly controlled, and that its overexpression and/or prolonged expression during S/M phase causes extra cell divisions and a small-cell phenotype.

## Discussion

Here we have made use of the uniquely accessible mitotic size control pathway of *Chlamydomonas reinhardtii* to identify a new cell-size regulator, CDKG1, whose activity enables mother cells to execute the correct number of mitotic cell divisions according to mother cell-size. Key properties of CDKG1 indicate that it couples mother cell size to the extent of cell division during S/M (*Figure 7*). Loss of function (*Figure 1*) and mis-expression phenotypes (*Figure 6*) show that while dispensable for overall cell cycle progression, CDKG1 is limiting for the normal number of S/M division cycles of

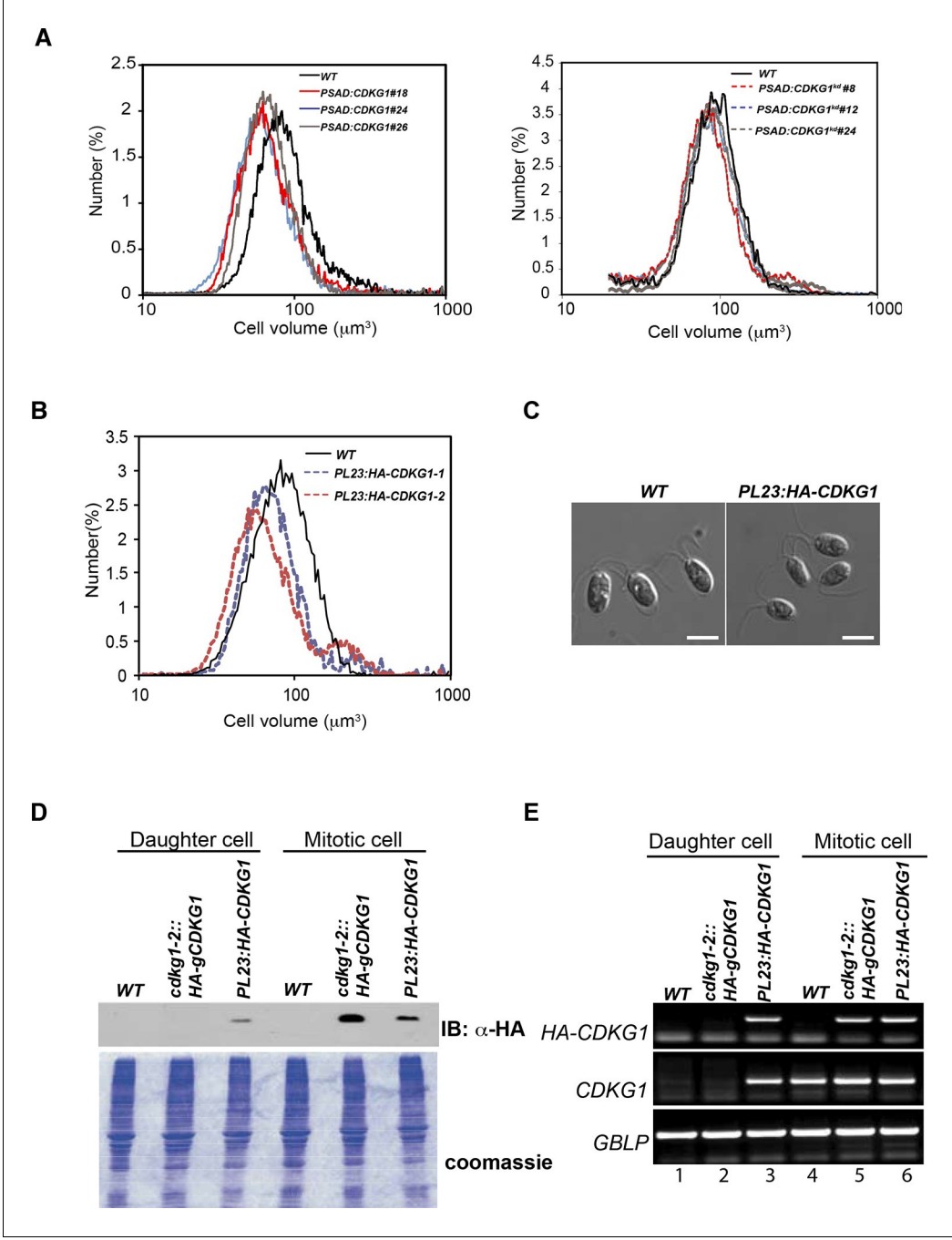

**Figure 6.** Mis-expression of *CDKG1* causes a small-cell phenotype. (**A**) Daughter cell size distributions of three independent *PSAD:CDKG1*-expressing transformants along with a wild type (WT) control (left panel); daughter cell size distributions of three independent *PSAD:CDKG1^kd^*-expressing transformants (kd: kinase dead) and a wild type (WT) control (right panel). (**B**) Daughter cell size distributions of two independent *PL23:HA-CDKG1* transformants with wild type (WT) control. (**C**) DIC images of wild type (WT) daughter cells and daughters from *PL23:HA-CDKG1* transformants. Scale bar=7.5 μm. (**D**) Anti-HA immunoblot (IB) detection of HA-CDKG1 with total protein extracts from synchronized strains as indicated (upper panel). Coomassie stained gel after membrane transfer serves as a loading control (lower panel). (**E**) RT-PCR detection of the *HA-CDKG1* and/or endogenous *CDKG1* message from indicated strains: *PL23:HA-CDKG1* (lane 3 and 6), *cdkg1-2::HA-gCDKG1* (lane 2 and 5) and an untransformed parental control strain (lane 1 and 4) using RNA from daughter cells or S/M phase cells. *GBLP* is an internal control.

*Figure 6 continued on next page*

*Figure 6 continued*

The following figure supplements are available for figure 6:

**Figure supplement 1.** Cell division numbers of *PSAD:CDKG1* expressing cells Unsynchronized wild type and isogenic *PSAD:CDKG1* cells were plated on HSM agar in the dark and division numbers were scored by percentage in each category: 0—no division (pre-Commitment), 1—one division (two daughters), 2—two divisions (four daughters), and 3—three divisions (eight daughters).

**Figure supplement 2.** Loss of expression and size phenotypes of *PL23:HA-CDKG1* strains after repeated passaging, and normal size phenotypes of strains expressing *PL23:HA-CDKG1^{kd}*.

mother cells and thereby acts as a determinant of daughter cell-size. The timing of CDKG1 expression is also consistent with its function as a size regulator during S/M phase: *CDKG1* message and protein accumulate in late G1 just before entry into S/M phase (*Figure 4A*, *Figure 3—figure supplement 1B* and *Figure 7A*); nuclear CDKG1 is then diluted with each round of division until the end of S/M when it disappears in daughter cells (*Figures 5A,E*, *Figure 5—figure supplement 1B* and *Figure 7*). The loss of CDKG1 does not eliminate size control completely, but instead alters the normal set points that govern the relationship between mother cell size and cell division number. Thus, other mechanisms must exist that trigger exit from S/M in the absence of CDKG1, but do so prematurely. *cdkg1* mutant strains may be valuable for investigating this next layer of size control that operates when CDKG1 is missing.

In principle, the level of a size regulatory protein such as CDKG1 might scale linearly with mother cell size; however, instead we observed an allometric scaling relationship where large mother cells produce a higher total concentration of CDKG1 protein and mRNA than smaller mother cells (*Figure 4G*, lanes 5, 6 versus 3, 4, *Figure 4—figure supplement 1*). Interestingly, *CDKG1* abundance was at least partly controlled by its 3′ UTR that conferred a cell-cycle-specific expression pattern to a heterologous reporter gene similar to the expression pattern of endogenous *CDKG1* mRNA (*Figure 4C*). We speculate that the unusually long and uridine-rich 3′ UTR of *CDKG1* plays an essential role in CDKG1 accumulation in mother cells, likely through the control of mRNA stability.

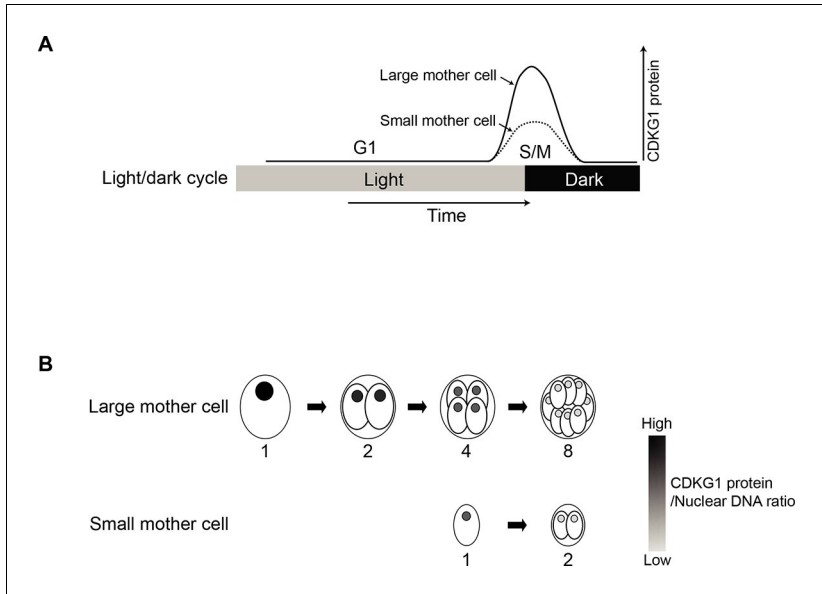

**Figure 7.** Model for CDKG1 as a limiting size regulator for cell division. (**A**) Schematic of cell cycle with kinetics of CDKG1 accumulation depicted by solid or dashed curves. Large mother cells accumulate higher concentrations of CDKG1 than small mother cells. (**B**) Nuclear CDKG1 to nuclear DNA ratio (gray scale) may act as a titratable regulator of cell division. The nuclear concentration of CDKG1 is reduced with each division. When it falls below a critical threshold cells exit S/M.

Control of mRNA stability of cell cycle regulators is an important mechanism operating in early embryonic development at the midblastula transition (*Tadros and Lipshitz, 2009*), but less is known how mRNA stability impacts somatic cell cycles where mRNAs for cyclins and CDKs such as the *CCND1* and *CDK6* are subject to cell-cycle-stage dependent differential turnover (*Deshpande et al., 2009*; *Sun et al., 2008*). Understanding how the 3' UTR of *CDKG1* confers a highly specific S/M phase expression pattern to its mRNA may shed light on how mRNA abundance is coupled to cell cycle progression in other contexts. Finally, we emphasize that mRNA abundance alone cannot explain the tightly controlled pattern of protein accumulation and degradation for CDKG1 where its net synthesis is restricted to a narrow window of time prior to S/M, its levels drop slowly during division, and it is completely eliminated in post-mitotic cells.

How might the abundance and turnover of CDKG1 mRNA and protein contribute to cell size control? A straightforward way of conceptualizing a sizer protein involves its synthesis in proportion to cell growth followed by triggering of a downstream cell cycle event when enough protein accumulated (*Fantes et al., 1975*). The G1 cyclin CLN3 in budding yeast has been proposed to function in this manner with its abundance being sensed through occupancy of nuclear DNA binding sites (*Wang et al., 2009*). However, a mitotic 'sizer' in Chlamydomonas cannot simply trigger a single event, but must instead provide quantitative information so that mother cells execute the correct number of cell divisions. Cell division counting mechanisms exist in early embryonic cell cycles of some metazoans where a fixed number of rapid division cycles occurs prior to the midblastula transition that is triggered when the cytoplasmic to nuclear DNA ratio (N:C) reaches a specific value (*Tadros and Lipshitz, 2009*). The ratio of CDKG1 to nuclear DNA changes dynamically during S/M as shown in *Figure 5E*, so a direct N:C sensing mechanism is possible. If a threshold amount of CDKG1 is required to initiate a new round of S/M, and the amount per nucleus decreases by over half with each division, then at some point the level of CDKG1 would fall below the critical amount. This model assumes that the main mode of reduction for CDKG1 is by dilution, but we also see some degradation (*Figure 5D*) and possibly there is inactivation of a portion of CDKG1 with each round of S/M. Either of these latter two effects might account for larger mother cells needing proportionally more CDKG1 than smaller mother cells. For example a small mother cell undergoing one cycle of S/M might require *one* unit of CDKG1 (one division, two daughters); while a four-fold larger mother cell would require *seven* units of CDKG1 for executing a total of *seven* mitoses (1+2+4) to produce eight daughters. Alternatively, the allometric scaling of CDKG1 abundance might reflect a signal amplification process that increases the robustness of the system output (cell division number) by buffering what might otherwise be noisy fluctuations in CDKG1 levels (*Di Talia et al., 2007*). Cell-size dependent effects on cell physiology are difficult to investigate, but do occur in other systems, such as yeast, where transcription can be size controlled (*Wu et al., 2010*), and in animals, where bulk protein turnover rates appear to be cell-size dependent (*Conlon et al., 2001*). Intriguingly, cell size is linked to developmental fate in the multicellular relative of Chlamydomonas, *Volvox carteri* where large post-mitotic daughters (>8 μm diameter) differentiate into reproductive cells and small post-mitotic daughters (<8 μm diameter) differentiate into sterile somatic cells (*Kirk et al., 1993*). We speculate that this size-dependent differentiation may be driven by a modification of the CDKG1-D cyclin-MAT3/RBR pathway that is conserved in *Volvox* (*Prochnik et al., 2010*). Finally, our data do not completely rule out more complicated and alternative models where other properties of CDKG1 besides its abundance, such as activation and inactivation by posttranslational modification or cyclin association, also contribute to its function in coupling mother cell size to cell division number.

Our model of allometric CDKG1 production and dilution can be compared to a recent model for size control in budding yeast where dilution of a negative cell cycle regulator, Whi5, provides a threshold triggering passage through Start that initiates the G1/S transition (*Schmoller et al., 2015*). In the former case a positive regulator is titrated away to control cell cycle exit, while in the latter it is a negative regulator being titrated away to control cell cycle entry; but in both models a fixed amount of a limiting factor is being used to gauge an important cellular change such as DNA content or cell volume. Non-linear scaling relationships between regulatory proteins and other cell constituents may be a general way of converting continuously scaling features of cells such as size to digital behaviors such as cell division that require an all or none response.

In yeast and mammalian cells, cyclin accumulation is typically rate-limiting for cell cycle progression (*Morgan, 1997*), but here we demonstrate that abundance of the CDK subunit, CDKG1, may

control the extent of cell division. Similarly, overexpression of the human RB kinase CDK6 is associated with sporadic cases of lymphoma (*Corcoran et al., 1999*; *Hayette et al., 2003*). Consistent with its role in regulating cell size and S/M phase entry upstream of MAT3/RBR, CDKG1 can directly phosphorylate RB/MAT3, and the simplest interpretation of our data is that the extent or rate of MAT3/RBR phosphorylation by CDKG1 on specific sites controls S/M phase entry. Although we could not detect changes in bulk phosphorylation patterns in MAT3/RBR caused by loss of CDKG1 (*Figure 3—figure supplement 2*) this is not inconsistent with CDKG1 being a MAT3/ RBR kinase. Animal RBRs are phosphorylated on over a dozen sites by multiple CDK-cyclin complexes with site specificity, redundancy and function of individual phosphorylation sites still not completely understood (*Rubin, 2013*). MAT3/RBR has 16 predicted CDK phosphorylation sites (*Umen and Goodenough, 2001*), any one of which might be a substrate for one or more CDKs and phosphatases. Indeed, at least three CDKs are likely to be active during S/M—CDKA1, CDKB1 and CDKG1 (*Tulin and Cross, 2014*), with the first two having mRNAs that are expressed at several fold higher levels than that of *CDKG1* (*Zones et al., 2015*). We predict that CDKG1 complexed with D-type cyclins will target a single or small subset of these sites to influence the activity of the MAT3/RBR complex towards promotion of cell division, an idea that can be tested when the phosphorylation sites and kinases for MAT3/RBR are better understood. While CDKG1 may have substrates other than MAT3/RBR, it appears to act mainly through MAT3/RBR in size control since the *cdkg1* mutation is completely suppressed in *cdkg1 mat3* double mutants. If there are additional CDKG1 substrates that influence the cell cycle, they either function through the MAT3/RBR-E2F1-DP1 complex or play a relatively minor role in size control.

Previous work on Chlamydomonas RB pathway mutants revealed defects in commitment threshold size as well as defects in S/M checkpoint suggesting that these two size control points are at least partially coupled. Here we have identified *cdkg1* as a mutant that appears to affect only the S/ M size control mechanism and not the size at which cells pass Commitment. Two lines of evidence support a specific role for CDKG1 in S/M size control. First, *cdkg1* mutants and *CDKG1* mis-expressing strains show daughter cell size phenotypes, but pass Commitment at close to wild-type size (*Figure 1*, *Figure 6*, *Table 1* and *Figure 1E*). Second, CDKG1 protein is produced in late G1 and S/M, and does not accumulate significantly at Commitment (*Figure 4A*, 4–6 hr). The identification of a separate S/M size control mechanism will enable future work aimed at dissecting the relationship between the Commitment and S/M sizers that both converge on the MAT3/RBR-E2F1-DP1 complex. It will also be interesting to determine how CDKG1 and the RB pathway connect to the NIMA related kinase, CNK2, that appears to function in both cell size and flagella length control (*Bradley and Quarmby, 2005*).

Yeasts have a single key cell cycle CDK (CDC28/cdc2) while metazoans have additional accessory CDKs such as CDK2, CDK3, CDK4 and CDK6 (*Morgan, 1997*) that have been proposed to be tied to the evolution of metazoan multicellularity (*Malumbres and Barbacid, 2009*). Our work with CDKG1 shows that unicellular eukaryotes also evolved complex cell cycle regulatory systems with multiple CDKs, and that this complexity was most likely lost in fungi rather than gained in metazoans. Moreover, the parallels between CDK4/6 and CDKG1 are striking: both are non-essential CDKs with variant PSTAIRE signature motifs; both bind specifically to and are activated by D-type cyclins; both phosphorylate RB-related proteins as their key substrates; and both are part of a system that integrates intra- or extra-cellular information (growth factors or cell size) into the cell cycle program (*Figure 2A,B* and *Figure 3—figure supplement 1F*). These similarities underscore the potential for Chlamydomonas to provide insights into cell cycle control mechanisms that evolved in ancestral single-celled eukaryotes prior to the divergence of major multicellular lineages.

## Materials and methods

### Strains, culture conditions and transformation

The following *Chlamydomonas reinhardtii* strains were obtained from the Chlamydomonas Stock Center (http://www.chlamy.org/strains.html): CC1690 (21gr wild-type *MT+*), CC1691 (6145c wild-type *MT-*), CC125 (*MT+*), CC124 (*MT-*), CC2453 (*nit1-305 MT-*), CC3995 (*mat3-4 nit1-305 MT+*). Strains were maintained on TAP agar plates and grown in liquid TAP for transformation and in HSM for synchrony and cell cycle experiments (*Harris, 1989*). Methods for culture growth,

synchronization, dark-shifting, and Commitment size measurement were described previously (*Fang et al., 2006*). Transformation of Chlamydomonas was done using either the glass bead method (*Kindle, 1990*) or by electroporation (*Shimogawara et al., 1998*) using a Biorad Genepulser Xcell (Bio-Rad, Hercules, CA). For electroporation the cell walls were not removed, the temperature during the pulse was 4°C, and the settings for a 0.4 cm cuvette were 800 kV, 25 µF, and no resistance.

## Insertional mutagenesis and screening for size mutants

Insertional mutagenesis with a *NIT1*-marked linearized plasmid pMN56 (*Nelson et al., 1994*) was used to generate tagged mutants in CC2453 (*nit1-305 MT-*) (*Tam and Lefebvre, 1995*). ~5000 individual Nit$^+$ transformants were screened directly in a Coulter Counter (Multisizer 3, Beckman Coulter, Brea, CA) for size defects. Positives were re-tested for linkage to the *NIT1* insertion by backcrossing to a *nit1-305 MT+* strain. Two large-cell mutations, *cdkg1-1* and *cdkg1-2*, were found to be linked to each other and subsequently found to have deletions in the *CDKG1* locus. The *cdkg1-1* and *cdkg1-2* insertion borders were mapped by adaptor ligation and PCR amplification using NIT1 insertion mapping primers listed in *Supplementary file 2* (*O'Malley et al., 2007*). Both alleles were found to have chromosomal deletions accompanying the *NIT1* insertion event, the smaller of which was in *cdkg1-2* (*Figure 1B* and *Figure 1—figure supplement 1A*).

## Complementation of *cdkg1*

A 7 kb fragment containing the full-length genomic region of *CDKG1* was amplified from BAC clone 36H9 using primers OER0003/OER0004 (http://www.chlamy.org/bac_details.html) and ligated into pBluescript SK- (Primers described in *Supplementary file 2*). A triple hemagglutinin tag (3XHA) was inserted just after the CDKG1 start codon using overlapping PCR. The resulting genomic fragment was ligated into pEZ (*Rasi et al., 2009*) at XbaI/EcoRI sites to replace the entire *PSAD* expression cassette to generate plasmid pEZ:HA-gCDKG1. pEZ:HA-gCDKG1 was transformed into CC124 by electroporation and transformants were selected on TAP with 20 µg/mL paromomycin and 10 µg/mL zeocin. Western blotting identified a primary transformant expressing HA-CDKG1 and this strain was crossed with *cdkg1-2* (*MT+*). Recombinant *cdkg1-2::HA-gCDKG1* progeny were found to be complemented for their size phenotype. Expression of the complementing transgene was verified using RT-PCR and Western blotting (*Figure 1D*, *Figure 1—figure supplement 1B and C*).

## Phylogenetic analysis of CDKs

CDK sequences (*Supplementary file 1*) were aligned and manually edited to remove gaps. Evolutionary models were determined using Prottest (*Abascal et al., 2005*) optimized with Akaike Information Criteria (AIC). Phylogenetic trees were constructed using the Maximum Likelihood method (PhyML) with approximate likelihood ratio (aLRT) support statistics (*Guindon and Gascuel, 2003*).

## cDNA cloning

All full-length cDNAs for Chlamydomonas CDKs and cyclins used in this study were amplified from wild-type strain CC1691 using RT-PCR with gene specific primer sets (*Supplementary file 2*), ligated into pGEM-T-easy (Promega, Madison, WI), and verified by sequencing.

## CDKG1 site-directed mutagenesis

CDKG1 kinase inactive (K125R) and T-loop inactivation (T254A) cDNA clones were generated using a QuickChange Site-Directed Mutagenesis Kit (Agilent Technologies, Santa Clara, CA). Primer sets for mutagenesis are listed in *Supplementary file 2*.

## CDKG1 antibody generation

pET28a-CDKG1 or pET28a-CDKG1ΔN (missing residues 1–92) were made by PCR amplification from pGEM-T-CDKG1 using primer sets described in *Supplementary file 2* and ligation into pET28a (EMD Millipore, Billerica, MA) at NdeI/EcoRI sites. pGST-MAT3 was generated as described previously (*Olson et al., 2010*). All recombinant proteins were expressed using *E.coli* BL21 codon plus-RIL strain (Agilent Technologies). Purification of insoluble 6xHis-CDKG1 and 6xHis-CDKG1ΔN was performed under denaturing conditions as described previously (*Olson et al., 2010*). Purified 6xHis-

CDKG1 was cut out from a Coomassie blue stained SDS-PAGE gel and sent to Cocalico Biological Inc. (Reamstown, PA) to generate rabbit polyclonal anti-sera. Polyclonal antibodies raised against CDKG1 were affinity purified with AminoLink Plus Resin (Pierce Thermo Fisher Scientific, Waltham, MA) coupled to purified His-CDKG1ΔN as described previously (*Olson et al., 2010*).

## Chlamydomonas whole cell extract preparation

50 to 100 mL samples from Chlamydomonas cultures at $5 \times 10^5$–$10^6$ cells mL$^{-1}$ were mixed with Tween-20 to a final concentration of 0.005% and collected by centrifugation at 3000 g for 5 min at 25°C. Pellets were resuspended in 500 µL lysis solution (1xPBS pH 7.4, 1x Sigma plant protease inhibitor cocktail (Sigma-Aldrich, St Louis, MO), 5 mM Na$_3$VO$_4$, 1 mM NaF, 1mM Benzamidine, 500 µM PMSF, 1 µM ALLN, 1µM MG-132), transferred to 1.5 mL tubes and centrifuged at 3000 g for 5 min at 25°C. Supernatants were discarded. Pellets were resuspended in lysis buffer to a final concentration of $5 \times 10^8$ cells mL$^{-1}$, and immediately frozen in liquid nitrogen. Pellets were thawed quickly on ice and centrifuged immediately at 20,000 g for 10 min at 4°C. Supernatants were mixed with NP-40 (0.1% final concentration) and the extract used immediately or frozen at -80°C. Protein concentration was determined using a Pierce$^{TM}$ BCA Protein Assay Kit (Thermo Fisher Scientific) with bovine serum albumin as a standard.

## Western blotting

Protein samples were subjected to standard SDS-PAGE (*Sambrook and Russell, 2001*) and transferred to PVDF membranes (Bio-Rad) in 25 mM Tris, 192 mM glycine, 20% methanol. Blots were blocked in PBST (1x PBS pH 7.4, 0.05% Tween-20) containing 5% non-fat dry milk for 1h at room temperature (RT) and incubated with primary antibodies diluted in PBST with 1% dry milk at 4°C overnight. Antibody dilutions were as follows: rat-anti-HA (1:2000) (3F10, Roche, Switzerland), rabbit-anti-Myc (1:1000) (SC-789, Santa Cruz Biotechnologies, Dallas, TX), rabbit anti-CDKG1 (1:500), rabbit anti-histone H3 (1:20,000) (PA5-16183, Thermo Fisher Scientific) and mouse anti-α-tubulin (1:40,000) (T6074, Sigma-Aldrich). After washing in PBST for three times 10 min, the blot was incubated with horseradish peroxidase (HRP) conjugated goat-anti-rat-IgG (1:5000) (Pierce ECL, Thermo Fisher Scientific) or goat-anti-rabbit-IgG (1:10,000) (Thermo Fisher Scientific) for 1h at RT, washed as described above, and processed for chemi-luminescence detection (Pierce ECL, EMD Millipore) using a Bio-Rad quantitative imaging system (Molecular Imager Chemi DocXRS+ Imaging System) or autoradiographic film (HyBlot CL).

## Yeast two hybrid assays

All yeast two-hybrid (Y2H) constructs were generated by subcloning cDNAs into either pGBKT7 (bait vector with GAL4 DB domain) or pGADT7 (prey vector with GAL4 AD domain) at NdeI/EcoRI sites (Clontech Laboratories, Mountain View, CA). Y2H methods were taken from the Clontech Yeast manual. All bait and prey constructs were co-transformed into yeast strain MaV203 (Thermo Fisher Scientific) and selected on –Trp/-Leu synthetic dropout agar plates. The interactions were tested for production of X-gal, uridine auxotrophy on -Trp/-Leu/-Ura dropout plates, or histidine auxotrophy on -His/-Trp/-Leu dropout agar plates with the addition of 10, 15, 20 and 25 mM 3-amino 1,2,3 triazole (3-AT).

## Yeast *cdc28* (*cdk1*) mutant complementation

For yeast *cdc-28* complementation, pVL399-CDKG1, pVL399-CDKG1$^{kd}$, and pVL399-CDKG1$^{ΔT\text{-}Loop}$ (T-loop inactivation) (*Figure 2—figure supplement 1*) were generated by inserting cDNAs into pVL399 (*Gao et al., 2007*) at the XhoI site. Plasmids were transformed into strain CWY181 *cdc28-13* (*bar1::ura3 cdc28-13 ade1 his2 leu2-3, 112 trp1-1 ura2*) (*Reed and Wittenberg, 1990*) with selection on SD –Leu agar plates. Growth at permissive (30°C) and non-permissive (37°C) temperatures was scored after two days.

## *In vitro* protein expression and GST pulldown assay

pGADT7-CDKG1, pGADT7-CDKG1$^{kd}$, pGBK-CYCD2 and pGBK-CYCD3 were used to express of HA-CDKG1, HA-CDKG1$^{kd}$ (kinase dead), myc-cyclin D3, and myc-cyclin D2 *in vitro* using TNT T7 Quick Coupled Transcription/Translation System with reticulocyte lysates (Promega) according to

the manufacturer's instructions. Each 20 µL of IVT product (HA-CYCD3, HA-CYCD2, HA-CDKG1 or HA-CDKG1-ΔN) was mixed with 30 µL glutathione sepharose beads (GE Healthcare, Pittsburgh, PA) bound with 2 µg GST-MAT3 or 2 µg GST, and the total volume was adjusted to 100 µL by adding binding buffer (1x PBS pH7.4, 0.1% NP-40, 1x Sigma plant protease inhibitors cocktail, 500 µM PMSF, 1 mM Benzamidine, 1mM NaF, 5 mM $Na_3VO_4$). After gentle mixing for 2 hr at 4°C, beads were thoroughly washed in binding buffer and bound proteins were eluted into SDS-PAGE sample buffer by boiling for 5 min, separated by SDS-PAGE, and detected by Western blotting with anti-HA antibodies as described above.

### *In vitro* kinase assay

Equal amounts of HA-CDKG1 or CDKG1[kd] (kinase dead) from IVT were immunoprecipitated with and without IVT myc-cyclin D3 or D2 products using anti-HA sepharose beads as above. After IP each 15 µL of sepharose beads was incubated in 20 µL kinase buffer (50 mM Tris-HCl pH 7.5, 10 mM $MgCl_2$, 10 µM ATP, 1 mM DTT, 500 µM PMSF, 1 mM Benzamidine, 1 mM NaF, 5 mM $Na_3VO_4$) with 10 µCi γ32P-ATP and 2.5 µg substrate protein: histone H1 (EMD-Millipore), purified GST-MAT3 or GST, respectively. Reactions were carried out at room temperature for 30 min and stopped by the addition of SDS-PAGE sample buffer and boiling for 5 min. Mixtures were separated by SDS-PAGE and substrate proteins were visualized by Coomassie blue staining. Incorporation of [32P] phosphate was measured with a phosphorimager (Molecular Dynamics Typhoon 8600, GE Healthcare). The kinase activity was calculated relative to the reaction with CDKG1 and GST-MAT3.

### *In vitro* co-immunoprecipitation

10 µl of HA-CDKG1 or HA-CDKG1[kd] IVT products were premixed with 10 µL myc-CYCD3 or myc-CYCD2 IVT products for 1h at 4°C then diluted to 50 µL with binding buffer (1x PBS pH7.4, 0.1% NP-40, 1x plant protein inhibitors cocktail (Sigma-Aldrich), 500 µM PMSF, 1 mM Benzamidine, 1 mM NaF, 5 mM $Na_3VO_4$), and mixed with 10 µL anti-HA sepharose beads (Roche) for 1h at 4°C. The beads were washed for 4 x 15 min in binding buffer and eluted with SDS-PAGE sample buffer by boiling for 5 min. The same proportion of input, unbound and co-IP products were subject to SDS-PAGE and Western blotting using anti-Myc or anti-HA antibodies.

### Immunoprecipitation of HA-CDKG1 and HA-CDKG1[kd] and kinase assay

Plasmid *pEZ:HA-CDKG1[kd]*, was generated by ligating a N-terminal 3xHA tagged CDKG1[kd] cDNA into the vector pEZ (*Rasi et al 2009*) and transformed into wild-type Chlamydomonas strain CC125. Transformants expressing HA-CDKG1[kd] were identified by Western blotting with anti-HA antibodies. Synchronized *cdkg1-2::HA-gCDKG1* and *HA-CDKG1[kd]*cells were collected at S/M phase and subjected to IP with anti-HA sepharose beads (Roche). After IP, equal volumes (15 µL) of IP beads from each sample were used as a source of kinase activity to perform kinase assays with histone H1, purified GST-MAT3 and GST as described above or were subjected to SDS-PAGE and Western blotting to detect HA-tagged protein.

### Co-immunoprecipitation of HA-MAT3 and CDKG1

*mat3-4::HA-MAT3* cells that express a complementing HA-tagged MAT3 (*Olson et al., 2010*) construct were synchronized in a 14 h:10 hr light-dark cycle. At each cell cycle stage samples were collected and used to generate whole cell lysates that were subjected to IP with anti-HA sepharose beads (Roche) as described previously (*Olson et al., 2010*). Bound proteins were eluted, separated by SDS-PAGE, and detected by Western blotting as described above.

### Immunofluorescence staining and quantitation

Synchronized *cdkg1-2::HA-gCDKG1* or control strain CC1691 were fixed with 2% paraformaldehyde in PBSP solution (1x PBS pH7.4, 1 mM DTT, 1x plant protease inhibitor cocktail (Sigma-Aldrich), 1 µM ALLN and 1 µM MG-132) for 30 min on ice. Fixed cells were adhered to polylysine-coated cover slips and extracted in cold methanol 2 x 5 min at -20°C, and rehydrated in 1 mL PBSP 30 min at 4°C. After rehydration slides were blocked for 30 min in blocking solution I (5% BSA and 1% cold water fish gelatin (Sigma-Aldrich) in PBSP) and blocking solution II (10% goat serum, 90% blocking solution I). Coverslips were incubated with α-HA monoclonal antibodies (Roche 3F10) (1:1000 dilution in 20%

blocking solution I) at 4°C for 12 hr. Coverslips were washed 3 x10 min in 1% blocking solution I, and then incubated with Alexa Fluor 488 conjugated goat anti-mouse IgG (1:1000) ( Thermo Fisher Scientific) in 20% blocking solution I) for 1 hr and then incubated with DAPI (5 µg/mL in $H_2O$) for 5 min. Stained cells were washed in 1 x PBS for 3 x 3 min and mounted with Mowiol 4–88 (Polysciences Inc, ) containing 0.01% PPD (p-phenylenediamine). Z-stacks of fluorescence images were collected with a Zeiss 710 confocal microscope (Zeiss, Germany) equipped with a 488 nm argon laser (for Alex Fluor 488) and 405 nm UV laser (for DAPI). Quantification of HA-CDKG1 immunofluorescence intensity was done using ImageJ (*Abramoff et al., 2004*). For quantitation of IF signals cells identified first from DIC images and data from all cells in a field were collected. Identical numbers of Z stacks for each cell were collected with identical settings (using untagged control samples to determine background) and converted to maximum projection images that were imported into ImageJ for further analysis. Cell boundaries were defined manually using the DIC image. Nuclear area was defined by automated segmentation of the DAPI signal. The total nuclear HA-CDKG1 immunofluorescence intensity was calculated as the portion of the 488nm signal that overlapped with the DAPI stained region of the cell. The nuclear area (A) was converted to the nuclear volume (V) using a formula that approximates nuclei as spheres $V = 4/3*(A/\pi)^{3/2}$. Nuclear concentrations of HA-CDKG1 were calculated as nuclear HA-CDKG1 immunofluorescence intensity divided by nuclear volume (V). Cells where at least three divisions were predicted to occur based mother size were used in order to allow meaningful comparisons of CDKG1 concentrations at different division numbers. Mean values of nuclear concentration of HA-CDKG1 in daughter cells were compared between dividing cells at different stages (2, 4, or 8 nuclei) using a non-parametric two-tailed t test (*Figure 5D*). The ratio of HA-CDKG1 to nuclear DNA was calculated as total nuclear HA-CDKG1 immunofluorescence intensity summed over all nuclei within the mother cell divided by nuclei number within the same mother cell, considering one nucleus as one copy of total nuclear DNA.

## Nucleus:cell volume (N/C) ratio measurements

The plasmid, pMF124cGFP (*Fuhrmann et al., 1999*) was transformed into wild-type strain 21gr by electroporation (see above) and selected on TAP agar with 20 µg/mL zeocin to generate *ble-GFP* strains. Transformants were grown and re-screened on up to 200 µg/mL Zeocin followed by live cell fluorescence imaging to identify transformants with stable, bright nuclear GFP fluorescence. A *ble-GFP* strain with strong GFP signal was cultured in HSM medium at 25°C and synchronized under a 12 hr light:12 hr dark cycle with bubbling aeration of air and 0.5% $CO_2$. Culture illumination was provided by a combination of red (625 nm) and blue (465 nm) LEDs each set for 150 µE for a total of 300 µE. Samples were collected during the peak of S/M phase between 12 hr light and 1 hr dark. Cells were suspended in 2% low melting temperature agarose and immediately mounted on a glass slide with a coverslip and sealed with clear nail polish. Cells were imaged with a Leica DMI 6000 B (Leica, Germany) wide field fluorescence microscope equipped with a 100x oil objective lens (NA 1.40), Photometrics Coolsnap HQ2 CCD camera and a metal halide light source (Sutter Instruments, Novato, CA) using a standard FITC/GFP filter set (Leica L5). 0.2 µm offset Z stacks of green fluorescence and maximum diameter DIC images were captured for each cell. A maximum projection image of the GFP fluorescence signal was used to visualize nuclei. Nuclei were approximated as spheres with diameters (d) used to calculate nuclear volume (V) by the formula $V = 1/6\pi d^3$. Individual DIC images focused at the midpoint of each cell were used to measure cell volume that was approximated as an ellipsoid with major (a) and minor (b) axes. Cell volume (V) was calculated as previously described using the equation $V = \pi ab^2/6$ (*Umen and Goodenough, 2001*). All images were processed with FIJI software (*Schindelin et al., 2012*). N/C ratios in dividing cells were grouped according to number of nuclei within a mother cell (1,2,4,8,16 nuclei). Data were collected from at least 24 cells per group, except for the 16-cell group (n=11). No significant differences were detected among groups using a one-way ANOVA test (p=0.09879, d=4).

## Luciferase assay

All *Gaussia princeps* luciferase expression constructs were generated from pHsp70A/Rbcs2-cgLuc (*AR:luc:RBCS*) obtained from the Chlamydomonas Stock Center (http://chlamycollection.org/plasmids/) by exchanging the promoter-5′UTR (HSP70A-RBCS2) or the terminator-3′UTR (RBCS2) sequences that flank the *C. reinhardtii* codon-adapted luciferase gene *luc* (*Ruecker et al., 2008*).

*AR:luc:RBCS* also contains a lox recombination site that allows site-specific-recombination-based introduction of selectable markers as described below (*Heitzer and Zschoernig, 2007*). For construct *AR:luc:G1*, the 1.7 kb *CDKG1* 3' untranslated region was amplified with primers CDKG1 3'UTR fw (Bgl II) and CDKG1 3'UTR rv (NcoI) (*Supplementary file 2*), cut with the indicated restriction enzymes, and ligated into *AR:luc:RBCS* cut with *BamHI-NcoI* sites to replace the RBCS2 3'UTR. To make pCDKG1-luc-RBCS (*G1:luc:RBCS*) and pCDKG1-luc-G1 (*G1:luc:G1*), a 746 bp fragment upstream of the CDKG1 start codon was amplified with primers gCDKG1 pro 1F (XbaI) and gCDKG1 pro 1R (XhoI), cut with the indicated enzymes, and ligated into XbaI-XhoI digested *AR:luc:RBCS* or *AR:luc:G1* to replace the AR promoter and 5' UTR with that of *CDKG1*. *AR:luc:RBCS, AR:luc:G1*, and *G1:luc:G1* were recombined *in vitro* with pKS-aph7-lox obtained from the Chlamydomonas Stock Center (http://chlamycollection.org/plasmids/) using Cre-recombinase (New England Biolabs, ) as described in (*Heitzer and Zschoernig, 2007*) to introduce the selectable marker *AphVII* that confers hygromycin resistance.

All luciferase expression constructs were transformed into *Chlamydomonas* wild-type strain 6145c (*MT-*) and selected on TAP agar plates with 20 µg/mL hygromycin. Individual transformants were grown to mid-log phase in 200 µL of liquid TAP medium in 96-well microtiter plates and screened for luciferase expression by luminescence. For time course experiments, strains were synchronized as described above for N:C ratio experiments. 1 mL of cells was pelleted every 3h after addition of Tween-20 to 0.005%. Pellets were resuspended in HSM and immediately frozen in liquid $N_2$. After thawing, samples with equal amount of biomass (cell number x mean cell volume) were loaded in 96-well opaque white plates (PerkinElmer Cat.No. 6005290, Waltham, MA) and resuspended in 100 µL HSM. Luminescence was measured with a BMG Fluostar Optima luminometer with a gain setting of 4095 and an interval time of 5 s after autoinjection of 100 µL substrate into each well. The luminescence substrate was 20µM coelenterazine (Gold Biotechnology, ) in 100 mM Tris pH 7.5, 500 mM NaCl, and 10 mM EDTA. Background was determined by measuring wells filled with HSM only and wells with untransformed cells. After subtracting background all luminescence values were normalized by biomass (total cell number x mean cell volume). Units of luciferase activity during the diurnal cycle were plotted relative to the 0h sample that was set to 1 (*Figure 4B*).

## CDKG1 mis-expression

*PSAD:CDKG1* was generated by inserting the full length *CDKG1* cDNA described above into the vector pEZ (*Rasi et al., 2009*) at NdeI/EcoRI sites. *pEZ:CDKG1* or *pEZ:CDKG1$^{kd}$* were transformed into wild type strain CC125 by electroporation. Transformants were sequentially selected on TAP agar plates containing 20 µg/ml paromomycin and then 10 µg/ml zeocin. Expression of *CDKG1* or *CDKG1$^{kd}$* was confirmed by RT-PCR with trans-gene specific primers (CDKG1 8F/ pSAD 3'UTR) listed in *Supplementary file 2*.

To generate *pL23:HA-CDKG1*, the HA-CDKG1 fragment was amplified with primers SalI 3HA F and BclI CDKG1 R (*Supplementary file 2*) from *pEZ:HA-CDKG1* in which a triple HA tag was inserted right after the start codon of a full length *CDKG1* cDNA coding region. The amplified HA-CDKG1 fragment was used to replace the luciferase gene between the BamHI and XhoI sites in vector *L23:luc:L23* (López-Paz et al., 2016 in preparation) a derivative of *AR:luc:RBCS* in which expression of luciferase is driven by ribosomal protein L23 (Cre04.g211800) promoter-5' UTR (1004 bp upstream of ATG) and terminator-3' UTR (824 bp downstream of stop codon). *pL23:HA-CDKG1* was recombined with plasmid *pKS-aph7-lox* as described above (*Heitzer and Zschoernig, 2007*) to generate *pAp7-L23:HA-CDKG1* that adds the selectable marker AphVII conferring hygromycin resistance to the construct. *pAp7-L23:HA-CDKG1* or *pAp7-L23:HA-CDKG1$^{kd}$* was transformed into wild-type strain 21gr by electroporation. Transformants were selected on TAP agar plates containing 30 µg/mL hygromycin. ~400 Hyg+ transformants were further screened in a Coulter Counter (Beckman Coulter Multisizer 3) for cell size phenotypes. *PL23:HA-CDKG1* or *PL23:HA-CDKG1$^{kd}$* strains were cultured in HSM medium at 25°C and synchronized under a 13 hr light:11 hr dark cycle but otherwise as described above for N:C ratio experiments.

RT-PCR to detect *HA-CDKG1* mRNA in *PL23:HA-CDKG1* strains was done as described previously (*Fang et al., 2006*). 3 µg RNA per sample was reverse transcribed with oligo dT and random hexamer (9:1), along with ThermoScript™ Reverse Transcriptase III (Thermo Fisher Scientific) following the kit instructions. PCR conditions and primers are listed in *Supplementary file 2*. GBLP (Genbank X53574.1) was used as an internal control.

For detection of HA-CDKG1 in *PL23:HA-CDKG1* or *PL23:CDKG1^{kd}* strains, whole cell extracts from synchronized cells at daughter and mitotic stages were prepared as described above, except using PBSP (1xPBS pH 7.4, 1x Biosciences protease inhibitor cocktail (, ), 1 mM PMSF, 10mM $Na_3VO_4$, 5mM NaF, 2 mM benzamidine, 2 µM ALLN, 2 µM MG132) as lysis solution. Cell extracts containing all detectable HA-CDKG1 were separated on 12% Tris-glycine SDS-PAGE gels (*Sambrook and Russell, 2001*), and Western blot detection was done as described in the previous section, except rat-anti-HA (1:3000) (Roche 3F10) was diluted in PBS with 3% dry milk.

### Quantitative Western blotting

Chemiluminescence signals were detected using a BioRad Chemi DocXRS+ Imaging System. Bio-Rad Image-Lab (4.1 PC version) software was used for signal measurement with the Volume Tool used to define each band along with a matched background volume that was subtracted. Total signal from all lanes of independent blots was used to normalize between blots.

### Phos-tag SDS-PAGE and immunoblot

*mat3-4::HA-MAT3* (*MT+*) (*Olson et al., 2010*) was crossed with *6145* (*MT-*)to generate *HA-MAT3* (*MT-*) progeny. *HA-MAT3* (*MT-*) was crossed to *cdkg1-2* (*MT+*) and *cdkg1-2::HA-MAT3* progeny were identified by genotyping (*Figure 3—figure supplement 2*).

Daughter cell and S/M stage lysates were prepared as described above for detection of *PL23: HA-CDKG1*. Lysates were de-salted using spin columns (Princeton Separation Centri-sep spin column, ), then mixed with 6x SDS-PAGE sample buffer. Protein samples were heated at 85°C for 5 min and centrifuged briefly prior to loading onto either normal SDS-PAGE gels or phos-tag SDS-PAGE gels prepared following the manufacturers instructions using 100 µM of Phos-Tag acrylamide and 100uM $MnCl_2$ in an 8% acrylamide gel (Wako Pure Chemical Industries Ltd, Japan) (*Kinoshita et al., 2009*) Phos-Tag SDS-PAGE gels were run in phos-tag running buffer (25mM Tris-base, 192mM glycine, and 0.1% w/v SDS, PH 8.3) at 80 V for 14 hr. After electrophoresis, phos-tag gels were incubated for 30 min in phos-tag transfer buffer (25 mM Tris, 192 mM glycine, 20% methanol, and 0.1% w/v SDS, 1 mM EDTA), washed twice for 30 min in phos-tag transfer buffer without EDTA, and then transferred to Immobilon-P PVDF membranes (EMD Millipore) with an Xcel-Blot apparatus (Thermo Fisher Scientific) at 50 V for 2 hr (*Kinoshita et al., 2009*; *Kinoshita-Kikuta et al., 2014*). Western blotting was done as described above for CDKG1 mis-expression experiments.

### Quantitative real-time RT-PCR

Total RNA was extracted from synchronized Chlamydomonas cells with Trizolreagent (Thermo Fisher Scientific) following the method of *Fang et al. (2006)* and treated with RNase-free Turbo DNase (Life Technologies). 4 µg total RNA was reverse transcribed with oligo dT and random hexamers (9:1) using ThermoScript Reverse Transcriptase (Thermo Fisher Scientific) according to the manufacturer's instructions at the following reaction temperatures 25°C 10 min, 42°C 10 min, 50°C 20 min, 55°C 20 min, 60°C 20 min, 85°C 5 min. SYBR-Green based qPCR reactions in triplicate were performed and quantitated in a Bio-Rad CFX96 system. Each 10 µL reaction contained 0.1 µL cDNA, 1x ExTaq buffer (Takara, Japan), 2 mM $MgCl_2$, 0.5x SYBR Green I (Molecular Probes, Thermo Fisher Scientific), 0.05% Tween 20, 0.05 mg/mL BSA, 5% DMSO, 200 µM dNTPs, 0.3 µM primers, and 10 units of Taq DNA polymerase. qPCR Primers for luciferase (*gluc*) and *CDKG1* are listed in *Supplementary file 2*. Expression was normalized against *GBLP* (Genbank X53574.1) or 18s rRNA (*Fang et al., 2006*) as internal controls. The melting curve was examined for each reaction to ensure that no primer dimers or non-specific PCR products were present.

## Acknowledgements

We thank Garrett Anderson for the initial mapping of the *cdkg1* insertion mutations. We thank Curt Wittenberg for yeast strain CWY181. We thank Marina Wantanabe, Wanda Waizenegger, Tuya Wulan, Fuqin Sun, Richard Davenport, Thomas Connell, Kerri Husa, and Nazifa Hoque for laboratory support; Kimberly Calhoun, Tristan Harris and Clarence Pasion for assistance with the cell-size screen; and Peter De Hoff for helpful discussions. We thank Dmitri Nusinow for use of the BioRad Chemi DocXRS+ Imaging System. This work was supported by American Cancer Society Research Scholar Grant RSG-05-

196-01-CCG and National Institutes of Health Grant GM092744 to JGU, and a National Research Service Award Fellowship GM086037 to BJSCO.

## Additional information

### Funding

| Funder | Grant reference number | Author |
| --- | --- | --- |
| American Cancer Society | RSG-05-196-01-CCG | James G Umen |
| National Institutes of Health | R01 GM069521 | James G Umen |

The funders had no role in study design, data collection and interpretation, or the decision to submit the work for publication.

### Author contributions

YL, Conception and design, Acquisition of data, Analysis and interpretation of data, Drafting or revising the article, Contributed unpublished essential data or reagents; DL, BJSCO, Acquisition of data, Analysis and interpretation of data, Drafting or revising the article; CL-P, Acquisition of data, Analysis and interpretation of data, Drafting or revising the article, Contributed unpublished essential data or reagents; JGU, Conception and design, Analysis and interpretation of data, Drafting or revising the article

### Author ORCIDs

James G Umen, http://orcid.org/0000-0003-4094-9045

## Additional files

### Supplementary files

• Supplementary file 1. CDK sequences used for phylogenetic analysis

• Supplementary file 2. Primers used in this study

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
