## [Decision Letter]

Thank you for submitting your work entitled "A new class of cyclin dependent kinase in *Chlamydomonas* couples cell size to cell division" for peer review at *eLife*. Your submission has been favorably evaluated by Richard Losick (Senior editor), a Reviewing editor, and three reviewers, one of whom is a member of our Board of Reviewing Editors.

The reviewers have discussed the reviews with one another and the Reviewing editor has drafted this decision to help you prepare a revised submission.

Summary:

The authors take advantage of the powerful genetic advantages of *Chlamydomonas* to identify an exciting new player in the mechanism that couples cell growth to cell division. The kinase CDKG1, a divergent CDK, was identified from a screen for cell-size mutants. *cdkg1* mutants exhibit relatively normal growth through G1 and commitment, but fail to undergo the normal number of S/M cycles, resulting in larger than normal cells. CDKG1 can interact with RB and might be in a complex with CYCD3 and genetically, acts upstream of RB. The authors show that CDKG1 is required for execution of the normal number of reductional cell divisions that determine cell size, and, over-expression of CDKG1 causes reduced cell size. They propose a model by which CDKG levels act as a "sizer" and that dilution + degradation of CDKG sets a limit on the number of cell divisions once the concentration of CDKG is below a threshold, divisions stop.

Size control is an important (and currently hot) topic in cell and developmental biology. This work includes exciting and significant findings, and the genetic and biochemical analysis of CDKG1 and its interaction with MAT3 is sufficiently convincing that the paper should make a major impact. All three reviewers felt that several issues need to be dealt with before publication, however.

Essential revisions:

1) The authors postulate that CDKG1 levels scale with cell size and that this is a unique property of CDKG1 that is important for its role in cell size control. One concern is how accurate the quantification of CDKG protein and nuclear size derived from images can be. A second concern is whether CDKG is unique in its accumulation vis-à-vis cell size. It is possible that many other proteins scale with cell size, so the accumulation of CDKG1 in large cells may not be unique or relevant to cell size control. For example, the mother cell could be stockpiling many different proteins necessary for the subsequent S/M divisions. So it would be helpful to look at a few other proteins in mother cells that could be stockpiled, such as a histone (e.g., MBT work in Amodeo, PNAS, 2015) plasma membrane protein, or nuclear envelope protein. If numerous other proteins show the same behavior as CDKG1, it would be hard to conclude which protein is actually being measured.

2) In the author's model, CDKG1 is a limiting factor for S/M divisions, such that gradual depletion of CDKG1 causes cessation of division at a specific size. However, this cannot be completely true because cells carrying a deletion of CDKG1 nevertheless undergo several divisions (Figure 1—figure supplement 1). If CDKG1 is limiting for cell division, how do cells completely lacking CDKG1 undergo division? The authors should address this issue in their discussion of the data. Also, Figure 1—figure supplement 1 is an important figure and should be moved out of the Supplemental Information.

3) How is [CDKG] actually measured in the cell to trigger a threshold effect on RB? What's the evidence that CDKB-phosphorylated RB acts in a dose dependent (or at least a threshold) manner to drive a cell cycle? This may be shown in the author's previous work, but if so, should be noted explicitly here.

4) The author's model is interesting and represents a reasonable interpretation of the data. Yet the data for the model are largely correlative. Also, the model relies, to a large extent, on the data in Figure 5, which suggest that the amount of CDKG1 in the nucleus decreases with each division. However, there is a significant amount of scatter in the data, and comparative measurements of total fluorescence are always difficult to carry out in a reliable and accurate manner. So the authors should do more to consider other models. One simple alternative is that CDKG1 promotes cell division, and upstream mechanisms that control the kinase activity of CDKG1, rather than its concentration, play a more important role in defining the number of cell divisions.

5) Finally, there is a great deal of data in this manuscript and some danger that side issues may detract from the main point. One example is the role of the CDKG1 3' UTR in its cell cycle regulation. While an interesting finding, the key point of the paper is that the amount of CDKG1 protein in the nucleus regulates cell division, probably decreasing as a result of cell division diluting the pool. The 3'UTR observation is not incorporated in a discussion of the mechanism of protein dilution (pure dilution, regulated degradation, other?). A tighter Discussion pulling together all of the germane observations, perhaps leaving related results for a subsequent paper would make for a more compelling discussion of the main point.

---

## [Author Response]

1) The authors postulate that CDKG1 levels scale with cell size and that this is a unique property of CDKG1 that is important for its role in cell size control. One concern is how accurate the quantification of CDKG protein and nuclear size derived from images can be. A second concern is whether CDKG is unique in its accumulation vis-à-vis cell size. It is possible that many other proteins scale with cell size, so the accumulation of CDKG1 in large cells may not be unique or relevant to cell size control. For example, the mother cell could be stockpiling many different proteins necessary for the subsequent S/M divisions. So it would be helpful to look at a few other proteins in mother cells that could be stockpiled, such as a histone (e.g., MBT work in Amodeo, PNAS, 2015) plasma membrane protein, or nuclear envelope protein. If numerous other proteins show the same behavior as CDKG1, it would be hard to conclude which protein is actually being measured.

The first concern above regards reliability of quantitation for the CDKG1 IF signal in Figure 5. We are aware that IF quantitation is potentially problematic and that care must be taken to eliminate subjective bias. For this reason we used a semi-automated method where all dividing cells were first identified and marked from a field of DIC images. IF data was collected from all of the pre-marked cells using an automated process of signal detection for the nucleus (DAPI channel) followed by quantitation of the overlapping CDKG1 signal. For this reason we are convinced that the reduction in CDKG1 IF signal in Figure 5 reflects a small but real reduction in its concentration as cells progress through each division. We also emphasize that the ratio of CDKG1 to nuclear DNA unambiguously drops by at least two-fold with each round of division and we argue that this ratio is likely to be the key parameter sensed in this system, though we acknowledge other possibilities as described in Point 4 below (paragraph three, Discussion).

The second concern relates to the uniqueness of the CDKG1 abundance scaling and whether there are other proteins that behave the same way. We disagree with the notion that what we observed for CDKG1 would have to be unique for it to be a useful gauge of mother cell size or regulator of division. It is completely conceivable that many proteins scale in abundance the same way as CDKG1, but CDKG1 was evolutionarily selected as a proxy to be used for division control. We also note that our bulk protein data from coomassie gels shows that most proteins do not scale the way CDKG1 does. However, as suggested by the reviewers, it was still worthwhile to examine the behavior of two other cell cycle regulated proteins, α-tubulin and histone H3. Transcripts of the former show cell cycle regulation with a peak around the time of division (Zones et al., 2015) while the histone transcripts and proteins were both previously measured and found to be cell cycle regulated (Waterborg et al., 1995). Our new quantitative Western blotting experiments in Figure 4—figure supplement 2 show that neither α tubulin nor histone H3 scaled the same way as CDKG1 in mother cells of different sizes and in non-dividing controls (paragraph two, subsection “CDKG1 production scales with mother-cell size”).

2) In the author's model, CDKG1 is a limiting factor for S/M divisions, such that gradual depletion of CDKG1 causes cessation of division at a specific size. However, this cannot be completely true because cells carrying a deletion of CDKG1 nevertheless undergo several divisions (Figure 1—figure supplement 1). If CDKG1 is limiting for cell division, how do cells completely lacking CDKG1 undergo division? The authors should address this issue in their discussion of the data. Also, Figure 1—figure supplement 1 is an important figure and should be moved out of the Supplemental Information.

We appreciate reviewers’ comments. This is an important point that we clarify in the first paragraph of the Discussion. In wild-type cells CDKG1 is limiting for executing the correct number of cell divisions (but not limiting in the sense of being required to do any division).

We agree about Figure 1—figure supplement 1 that has now been moved into Figure 1 as panel 1E.

3) How is [CDKG] actually measured in the cell to trigger a threshold effect on RB? What's the evidence that CDKB-phosphorylated RB acts in a dose dependent (or at least a threshold) manner to drive a cell cycle? This may be shown in the author's previous work, but if so, should be noted explicitly here.

This is a difficult question to answer as it requires a system with a quantitative readout of the effect of CDKG1 concentration on phosphorylation of RB which could be competing with other kinases, phosphatases and modifiers whose activities are likely integrated into a signal for cell cycle progression and/or for exit from S/M. We were able to address the question of whether phosphorylation of RB is strongly impacted by loss of CDKG1 (Figure 3—figure supplement 2) and found that it is not. This result is not surprising given that CDKG1 is non-essential and its mRNA is never expressed at more than a fraction of the levels of the two major cell cycle kinases, CDKA and CDKB. The possible roles of CDKG1 phosphorylation of RB are addressed in the revised Discussion section on paragraph five.

4) The author's model is interesting and represents a reasonable interpretation of the data. Yet the data for the model are largely correlative. Also, the model relies, to a large extent, on the data in Figure 5, which suggest that the amount of CDKG1 in the nucleus decreases with each division. However, there is a significant amount of scatter in the data, and comparative measurements of total fluorescence are always difficult to carry out in a reliable and accurate manner. So the authors should do more to consider other models. One simple alternative is that CDKG1 promotes cell division, and upstream mechanisms that control the kinase activity of CDKG1, rather than its concentration, play a more important role in defining the number of cell divisions.

We addressed the reliability of our IF data from Figure 5 in the response to Point 1 above.

Regarding alternative interpretations, our data showing that overexpression of CDKG1 leads to extra divisions is not correlative and suggests that cells are highly sensitive to the amount of this protein. Were cell-size control exerted mainly through an upstream regulator of CDKG1 then overexpression of CDKG1 would not have a phenotype. Our data on extra cell divisions in lines expressing CDKG1 from a strong promoter (Figure 6) and the overall sensitivity of cells to CDKG1 overexpression (paragraph two, subsection “Mis-expression of CDKG1 causes a small-cell phenotype”) lead us to conclude that levels of this protein are meaningful and tightly controlled, but acknowledge that other levels of control might also exist in the revised Discussion (paragraph three).

5) Finally, there is a great deal of data in this manuscript and some danger that side issues may detract from the main point. One example is the role of the CDKG1 3' UTR in its cell cycle regulation. While an interesting finding, the key point of the paper is that the amount of CDKG1 protein in the nucleus regulates cell division, probably decreasing as a result of cell division diluting the pool. The 3'UTR observation is not incorporated in a discussion of the mechanism of protein dilution (pure dilution, regulated degradation, other?). A tighter Discussion pulling together all of the germane observations, perhaps leaving related results for a subsequent paper would make for a more compelling discussion of the main point.

We understand this concern, yet at the same time felt that there were two important points to be addressed. One is the question how the overall expression pattern of CDKG1 might be controlled, and the second is how its abundance might be coupled to the extent of cell division. Our UTR experiments provide a straightforward answer to the first question suggesting that mRNA abundance is sufficient to explain the overall accumulation pattern of CDKG1 (though there must be additional control mechanisms that prevent continued accumulation during S/M and lead to its elimination in post-mitotic cells). We agree that this point was neglected in the Discussion and have added in a short section on these results (paragraph two).